# Associations in cell type-specific hydroxymethylation and transcriptional alterations of pediatric central nervous system tumors

Min Kyung Lee [1] ✉, Nasim Azizgolshani[1,2], Ze Zhang [1], Laurent Perreard[3], Fred W. Kolling [3], Lananh N. Nguyen[4], George J. Zanazzi[3,5], Lucas A. Salas [1] & Brock C. Christensen [1,6,7] ✉

Although intratumoral heterogeneity has been established in pediatric central nervous system tumors, epigenomic alterations at the cell type level have largely remained unresolved. To identify cell type-specific alterations to cytosine modifications in pediatric central nervous system tumors, we utilize a multi-omic approach that integrated bulk DNA cytosine modification data (methylation and hydroxymethylation) with both bulk and single-cell RNA-sequencing data. We demonstrate a large reduction in the scope of significantly differentially modified cytosines in tumors when accounting for tumor cell type composition. In the progenitor-like cell types of tumors, we identify a preponderance differential Cytosine-phosphate-Guanine site hydroxymethylation rather than methylation. Genes with differential hydroxymethylation, like histone deacetylase 4 and insulin-like growth factor 1 receptor, are associated with cell type-specific changes in gene expression in tumors. Our results highlight the importance of epigenomic alterations in the progenitor-like cell types and its role in cell type-specific transcriptional regulation in pediatric central nervous system tumors.

Central nervous system (CNS) tumors are the leading cause of cancer death in the pediatric population[1]. While major progress has been made in reducing the mortality in pediatric cancers in the past few decades, the magnitude of reduction in the mortality rate of CNS tumors have not been as substantial[2]. Even among patients who survive childhood cancers, those who have survived CNS tumors have the highest cumulative burden of disease post-survival[3]. Craniospinal radiation and neuro-toxic therapy are major risk factors for the future burden on quality of life with late effects including neurocognitive impairments such as academic and memory decline, and adverse health outcomes like abnormal hearing and growth hormone deficiency[4–9]. Efforts to address discrepancies in the reduction of mortality rates and extensive chronic health burdens later in life have been made with the recent advances in technology that have allowed for better insight into the molecular characterization of pediatric CNS tumors[10–22]. Molecular biomarkers are progressively being incorporated into the diagnosis and management of certain pediatric CNS tumor types[23].

[1]Department of Epidemiology, Geisel School of Medicine at Dartmouth, Lebanon, NH, USA. [2]Department of Surgery, Columbia University Medical Center, New York, NY, USA. [3]Dartmouth Cancer Center, Geisel School of Medicine at Dartmouth, Lebanon, NH, USA. [4]Department of Laboratory Medicine and Pathobiology, University of Toronto, Toronto, ON, Canada. [5]Department of Pathology and Laboratory Medicine, Geisel School of Medicine at Dartmouth, Lebanon, NH, USA. [6]Department of Molecular and Systems Biology, Geisel School of Medicine at Dartmouth, Lebanon, NH, USA. [7]Department of Community and Family Medicine, Geisel School of Medicine at Dartmouth, Lebanon, NH, USA. ✉e-mail: sarahminkyunglee@gmail.com; Brock.Christensen@dartmouth.edu

One method to supplementally diagnose and subtype CNS tumors is DNA methylation[24]. Capper et al. developed a classification method to address previous issues in inter-observer variability for histopathological diagnosis of many CNS tumors[24]. Since the development of this method, DNA methylation classification is now used regularly for certain pediatric CNS tumor types, like ependymomas, to understand the prognosis and manage treatment decisions[13,14]. This method utilizes bisulfite-treated DNA, which does not distinguish between 5-methylcytosine (5mC) and 5-hydroxymethylcytosine, although it has been indicated only 5mC signal from oxidative bisulfite-treated DNA alters the classification from this method[25,26]. Moreover, while advancements have improved management strategies for some tumor types, many other pediatric CNS tumor types remain underexplored.

DNA methylation is one of the most well-studied epigenomic marks, primarily known for its role in regulating gene expression. DNA methylation occurs when a methyl group is added to the 5-carbon position of a cytosine in the context of a Cytosine-phosphate-Guanine (CpG) dinucleotides by DNA methyltransferases (DNMTs)[27–32]. Methylation of CpG island promoters is associated with repression of gene expression while methylation of gene bodies is associated with activation of gene expression[33–35]. 5mC many times co-exist with H3K9me3 marks and do not overlap with H3K4me3 marks and H2A.Z[34,36,37]. In addition, DNA methylation marks function as genome stabilizers by silencing transposable elements[34,38]. The main ways DNA methylation is altered in cancer include genome-wide hypomethylation in repetitive elements like retrotransposable elements[39,40], hypermethylation of promoters[40–43], and propensity for cytosines in CpG contexts to be mutated[44–47].

Cytosines can also remain in a hydroxymethylated state (5-hydroxymethylcytosine, 5hmC). 5hmC is formed when 5mC is actively being demethylated by ten-eleven translocation (TET) enzymes[48–50]. TET enzymes add a hydroxyl group onto the methyl group to become 5-hydroxymethylcytosine, then add the hydroxyl group again to become 5-formylcytosine, then again to become 5-carboxylcytosine, which is excised to become unmethylated[48–51]. While 5hmC is an intermediate, it has been shown to have functional roles and be stable in the genome. Like 5mC, 5hmC has been associated with regulating transcription. It is enriched in gene bodies of active genes and in transcription start sites in which promoters are marked with H3K27me3 and H3K4me4[52,53]. 5hmC has also been shown to play roles in maintaining pluripotency and tumorigenesis[52,54]. While generally 5hmC levels are relatively much lower than 5mC levels, higher levels of 5hmC are found in the brain tissue compared to other tissue and in embryonal stem cells developmentally programmed neuronal cells[52,55–61]. Although progress has been made since the discovery of TET enzymes producing 5hmC[49–51], more investigation is needed to understand the functional roles of 5hmC. While alterations in hydroxymethylation patterns have not been as well examined, studies have indicated decreased hydroxymethylation across the genome in a variety of tumor types including adult and pediatric CNS tumors[26,54,62–70], and mutations in hydroxymethylation-associated genes such as *IDH1/2* and *TET1/2/3* have been associated with certain tumor types like gliomas and acute myeloid leukemia[62,71–74].

Numerous studies have established that brain tumors display intratumoral cellular heterogeneity[17,19,20,75–85]. While it is known that both DNA methylation and hydroxymethylation patterns are tissue type and cell type dependent[52,53,86–90], limited research has addressed cell type-specific DNA cytosine modification alterations in these tumors. This gap exists largely due to the high cost and limitations in technologies to profile cytosine modifications at the cell type-specific scale[91]. While the importance of cell type composition effects in epigenome-wide association studies has been well documented[92–96], single-cell methylation profiling strategies[97–100] are slowly developing in comparison to more accessible and commercially available genome profiling technologies focused on gene expression or chromatin accessibility. To address these shortcomings, computational methods have been developed to deconvolute cell type composition using DNA methylation for certain tissue types[91,101–109]. While these methods have greatly improved our understanding of the cell type composition effects on many epigenome-wide association studies, they have not been utilized in investigating cell type composition effects on brain tumors due to some limited applicability in brain tissue.

In this study, we use a multi-omic approach to study cell type-level epigenomic alterations in pediatric CNS tumors to maximize the applicability of currently available methods. By integrating single nuclei RNA-seq and cytosine modification data, we provide a more complete picture of the cytosine modification alterations associated with pediatric CNS types and cytosine modifications that are associated with changes in transcription at the cell type level in pediatric CNS tumors.

## Results
Our cohort included 32 tumor tissues (8 astrocytomas, 6 embryonal tumors, 10 ependymomas, 8 glioneuronal/neuronal tumors) and 2 non-tumor tissue (Table 1). To assess the potential normal tissue margin in our tissues that may confound downstream analyses, we first determined the tumor purity of our pediatric CNS tumor samples that were used to measure DNA cytosine modifications. Tumor purity in our samples varied but did not significantly differ based on tumor type or grade (Supplementary Fig. 1). The genetic variants associated with each tumor can be found in ref. 110.

### Epigenomic burden from altered cytosine modifications in pediatric CNS tumors
To determine the global epigenomic burden of altered cytosine modifications in pediatric CNS tumors compared to non-tumor pediatric brain tissue, we compared median beta values for both 5hmC and 5mC across samples at each CpG and determined the methylation dysregulation index (MDI). MDI is a summary measure of the epigenome-wide alteration of tumors compared to non-tumor tissue[111]. Tumor tissues ($N = 32$) displayed a decrease in median 5hmC beta values and a slight increase in median 5mC beta values compared to non-tumor tissue (Non-tumor tissue $N = 2$; KS test: 5mC: $D = 0.019$, $p < 2.2e{-}16$; 5hmC: $D = 0.19$, $p < 2.2e{-}16$; Fig. 1A). The 5hmC MDI values were not significantly different by tumor type ($N = 8$ (ATC), 6 (EMB), 10 (EPN), 8 (GNN)) or by tumor grade ($N = 14$ (G1), 5 (G2), 6 (G3), 6 (G4); Fig. 1B), whereas 5mC MDI values varied by tumor type. Embryonal tumors had the greatest extent of epigenome-wide alteration burden compared to non-tumor tissue, astrocytomas had the lowest burden of 5mC MDI compared to non-tumor tissue, and we observed increasing 5mC MDI with increasing tumor grade. 5hmC MDI and 5mC MDI were positively correlated ($R = 0.44$, $p = 0.013$, Fig. 1C). We repeated our analysis after removing one astrocytoma sample with an outlier 5hmC MDI value and observed consistent results (Supplementary Fig. 2). In addition, we determined MDI for distinct genomic contexts and again found consistent results in which 5mC MDI, but not 5hmC MDI values significantly varied among tumor types (Supplementary Fig. 3). Interestingly, both 5hmC MDI and 5mC MDI in gene body, enhancer and exon regions were slightly, but statistically significantly higher than 5hmC MDI and 5-MDI when adjusted for tumor types (Supplementary Table 2). For both 5hmC and 5mC, MDI were highest in enhancers, then gene body/ exon regions, and were lowest in promoter CpGs. We tested and confirmed that the burden of observed epigenomic alterations was not due to differences in tumor purity (Supplementary Fig. 4, Supplementary Table 3A). However, we did observe significant differences in 5mC MDI by tumor grade (Supplementary Table 3B). While 5hmC is prevalent at only 6% of 5mC, the level of dysregulation of the hydroxymethylome is comparable to the level of dysregulation of the methylome with 5hmC MDI being 49% of 5mC MDI (Supplementary Table 4). Our results

## Table 1 | Subject demographics

| | Tumor types | | | | | |
|---|---|---|---|---|---|---|
| | Total (*N* = 34) | Astrocytoma (*N* = 8) | Embryonal (*N* = 6) | Ependymoma (*N* = 10) | Glioneuronal/neuronal (*N* = 8) | Non-Tumor (*N* = 2) |
| **Sex** | | | | | | |
| F | 14 (41 %) | 3 (38%) | 3 (50%) | 5 (50%) | 1 (12%) | 2 (100%) |
| M | 20 (59 %) | 5 (62%) | 3 (50%) | 5 (50%) | 7 (88%) | 0 (0%) |
| **Age (years)** | | | | | | |
| Mean (SD) | 8.5 (±5.3) | 5.6 (±4.5) | 9.2 (±5.4) | 9.5 (±4.3) | 10.5 (±6.5) | 5.8 (±7.4) |
| **Grade** | | | | | | |
| High | 12 (35 %) | 0 (0%) | 6 (100%) | 5 (50%) | 1 (12%) | 0 (0%) |
| Low | 18 (53 %) | 8 (100%) | 0 (0%) | 4 (40%) | 6 (75%) | 0 (0%) |
| NEC/NOS | 2 (6 %) | 0 (0%) | 0 (0%) | 1 (10%) | 1 (12%) | 0 (0%) |
| Missing | 2 (5.9%) | 0 (0%) | 0 (0%) | 0 (0%) | 0 (0%) | 2 (100%) |
| **Location** | | | | | | |
| Metastasis | 1 (3 %) | 1 (12%) | 0 (0%) | 0 (0%) | 0 (0%) | 0 (0%) |
| Subtentorial | 19 (56 %) | 5 (62%) | 5 (83%) | 8 (80%) | 1 (12%) | 0 (0%) |
| Supratentorial | 14 (41 %) | 2 (25%) | 1 (17%) | 2 (20%) | 7 (88%) | 2 (100%) |

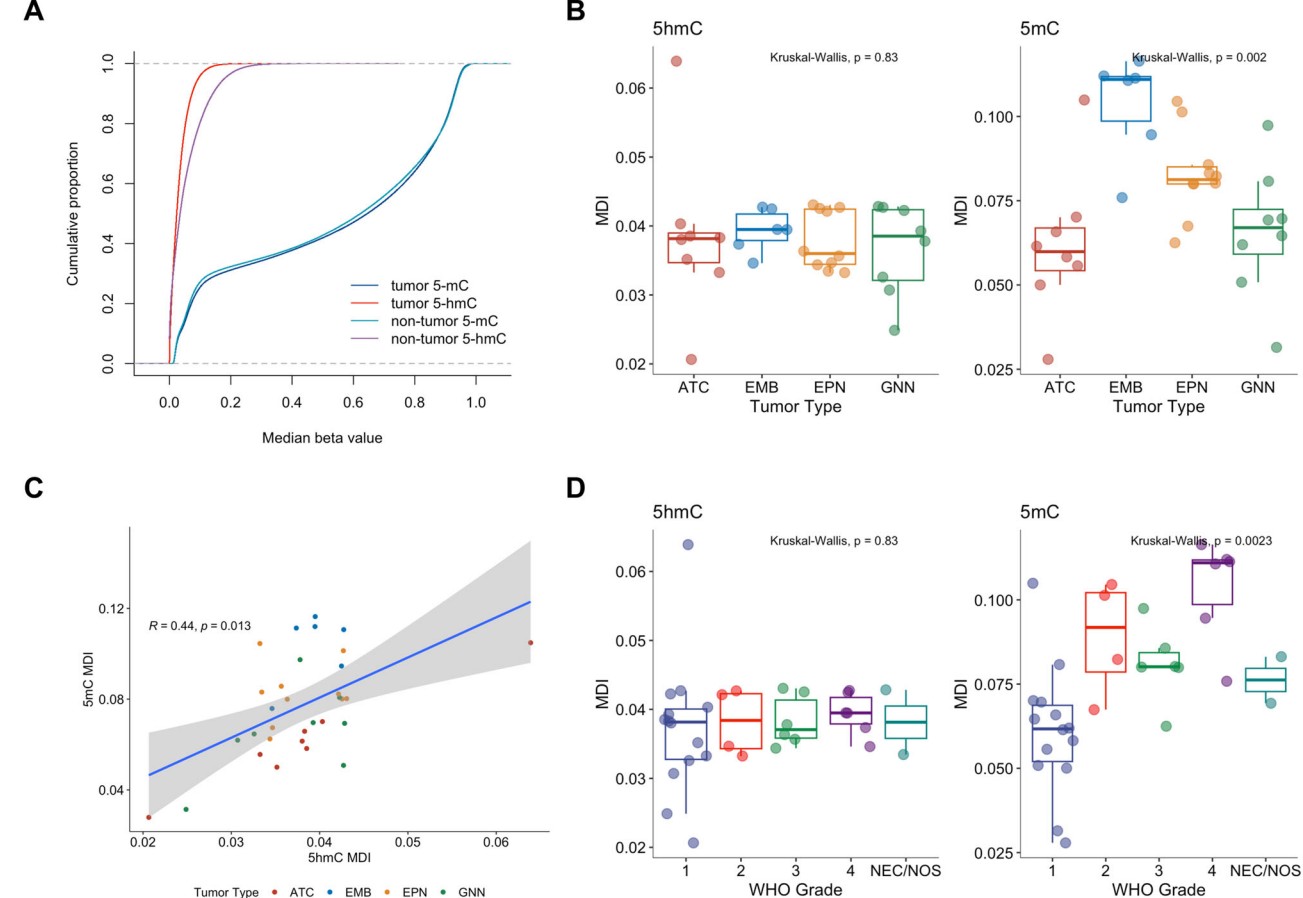

**Fig. 1 | Global methylation dysregulation, but not global hydroxymethylation dysregulation, is associated with tumor type and grade. A** Cumulative proportion of median 5hmC and median 5mC in tumors (*N* = 32) and non-tumor tissue (*N* = 2). **B** Methylation dysregulation index of 5-mC and 5mC by tumor type (*N* = 8 (ATC), 6 (EMB), 10 (EPN), 8 (GNN)) and (**D**) grade (*N* = 14 (G1), 5 (G2), 6 (G3), 6 (G4)). Differences in MDI were calculated using the rank-based Kruskal-Wallis test. **C** Correlation between 5hmC MDI and 5mC MDI calculated using Spearman rank

correlation. Linear regression line is indicated by the blue line. 95% confidence interval of the linear regression line indicated by gray bands. Color of each point indicates tumor type. In the boxplots of (**B**) and (**D**), the low ends of the segment indicate the minimum and the high ends of the segment indicate the maximum. Lower bounds of the box indicate the 25th percentile and the higher bounds of the box indicate the 75th percentile. Segment in the middle is the median. Source data are provided as a Source Data file.

suggest that while 5hmC may not be as prevalent, epigenome-wide alterations of 5hmC in tumors are occurring at comparable levels to altered 5mC.

## Cell type composition influences bulk-omics comparisons between pediatric CNS tumors and non-tumor pediatric brain tissue

We utilized our single nuclei RNA-seq data to identify the cell type composition of pediatric CNS tumor tissue and non-tumor pediatric brain tissue[110]. The cell types identified in our cohort like radial glial cells (RGC) in ependymoma were similar to comparable pediatric CNS tumors in previous literature[22,112]. As we wanted to account for major cell types present that may confound comparisons between the epigenomes of tumors and non-tumors, we identified the cell types present with most variability. Based on the cell type proportion distributions for all of our samples, we identified neuronal-like cells (NEU), neural stem cells (NSC), oligodendrocyte precursor cells (OPC), RGC, and unipolar brush cells (UBC) as having the most variance from PCA analysis (Supplementary Fig. 5A, B). For each tumor type we compared proportions of cell types with non-tumor pediatric brain tissue. Supporting our principal component analysis, the cell types with the greatest differences were NEU, NSC, OPC, RGC, and UBC (Supplementary Fig. 5C).

We conducted an epigenome-wide association study to determine the differential hydroxymethylated and methylated CpGs associated with each tumor type ($N = 8$ (ATC), 6 (EMB), 10 (EPN), 8 (GNN)) compared to non-tumor pediatric brain tissue ($N = 2$). To reduce potential confounding by cell type composition, we incorporated cell type proportions as covariates in a stepwise manner to each series of linear models. Age at diagnosis, sex, and tumor purity were adjusted to reduce potential confounding from these variables in these linear models. Due to sample size, tumor location was not included in the model. Importantly, as the number of cell type proportion covariates included in the models increased, the scope of differentially hydroxymethylated and differentially methylated CpGs associated with each tumor type decreased (Fig. 2A, Table 2, Supplementary Figs. 6–9, Supplementary Data 1–8). In addition, across our models in different tumor types, the extent of differentially hydroxymethylated CpGs (dhmCpGs) was far greater than that of differentially methylated CpGs (dmCpGs). When all five cell types (NEU, NSC, OPC, RGC, and UBC) were incorporated into the model, we observed low number of dmCpGs associated with each tumor type. Embryonal tumors had the greatest number of dhmCpGs, and the 83.1% were specific to the embryonal tumors (Fig. 2B). In the model with all five cell types included, 87 dhmCpGs were associated with astrocytoma, 850 dhmCpGs were associated with embryonal tumors, 31 dhmCpGs were associated with ependymoma, and 126 dhmCpGs were associated with glioneuronal/neuronal tumors. We identified 90 dhmCpGs (10.4%) that were shared across two or three of the tumor types and 28 dhmCpGs (3.2%) that were shared across all tumor types (Fig. 2B, Supplementary Table 5). The 28 shared CpGs were located predominantly in island (42.9%) and open sea (42.9%) regions in relation to CpG islands (Supplementary Table 6). In addition, 64.3% of the shared dhmCpGs were in DNase hypersensitive sites (DHS) (Supplementary Table 7). The shared CpGs tracked to genes including *ESRRG, HECA, THBD*, and *TJP1* (Supplementary Table 5).

We then investigated if specific genomic regions were associated with the changes in the number of dhmCpGs by assessing the relationship between the proportion of the dhmCpGs for each genomic context with each model using Spearman rank tests. We identified positive relationship between number of cell types included in the model and the proportion of dhmCpGs in regions within 200 bps of the transcription start sites (TSS200) and 1st exon regions (Fig. 2C, Supplementary Fig. 10). Moreover, we found negative relationship between the number of cell types included in the model and the

proportion of dhmCpGs in gene body, open chromatin, and transcription factor binding sites. Our results suggest that epigenome-wide association studies comparing bulk pediatric CNS tumor tissue to non-tumor pediatric tissue are considerably influenced by the cell type composition, especially in promoter and gene body genomic regions. Moreover, it was quite unexpected that the observed differences were almost solely in hydroxymethylation and not in methylation.

We then compared transcriptome data from bulk RNA-seq in each of the tumor types ($N = 8$ (ATC), 6 (EMB), 10 (EPN), 8 (GNN)) with non-tumor pediatric brain tissue ($N = 2$). The differential expression testing model included the same covariates (sex, age at diagnosis, and tumor purity) and the same five cell type proportions used for the EWAS analysis. Including proportions of major cell types of interest led to differences in an average of around 702 genes (range: 536–892) detected as significantly differentially expressed. In astrocytoma and glioneuronal/neuronal tumors, the adjusted model identified more genes that were significantly differentially expressed. In embryonal tumors and ependymomas, the adjusted model identified fewer genes that were significantly differentially expressed. Some key tumor progression-associated genes like *PTEN* in astrocytoma and in embryonal tumors, *MYCN* in ependymoma, and *BRCA2* in glioneuronal/neuronal tumors would not otherwise have been identified as significantly differentially expressed in the tumors had the cell type proportions not been adjusted for.

As we reduced potential confounding effects from cell type composition, we sought to explore genes with differential expression that were specifically associated with the tumors. Across all tumor types, the majority of differentially expressed genes were increased in expression compared to the non-tumor pediatric brain tissue (Supplementary Fig. 11A, 12–15, Supplementary Data 9–16). Almost half (43%, 3020 genes) of all genes with increased expression were shared across all tumor types (Supplementary Fig. 11B). Among the genes with shared increases in expression in tumors were *IRX5, MYOSLID, CWH43, ITGA2*, and *HOXA3*. Genes with increased expression across all tumor types were associated with biological oxidations and keratinization among other pathways (Supplementary Fig. 11D). There were 253 genes (13.6%) that had decreased expression shared across tumor types (Supplementary Fig. 11C), including *NPTXR, SCG2, B4GAT1*, and *ATRN*. Genes that were decreased in expression across all tumor types were associated with the insulin receptor signaling and ion channel transport among other pathways (Supplementary Fig. 11E). Our results suggest potential non-cell type-specific avenues for therapy that may be shared across the pediatric CNS tumor types.

To identify potentially important gene regulation by differential hydroxymethylation we compared changes in hydroxymethylation in dhmCpGs from the five-cell type-adjusted model with gene expression in each tumor type ($N = 8$ (ATC), 6 (EMB), 10 (EPN), 8 (GNN)). The genes we identified in our differential gene expression analysis were used in comparisons to changes in 5hmC. Generally, genes with decreased hydroxymethylation levels had increased gene expression across tumor types compared to non-tumor pediatric brain tissue (Fig. 3). When correlations between changes in 5hmC and changes in gene expression were performed to assess any directional relationship, the correlation coefficients across all tumor types were non-existent and not statistically significant even for genes that had statistically significant changes in gene expression ($R$, $p = -0.03$, 0.93 (ATC); $-0.02$, 0.85 (EMB); 0.096, 0.86 (EPN); 0.39, 0.19 (GNN), Fig. 3).

Only one dhmCpGs associated with ependymoma had significant decreased expression. The dhmCpGs with differential expression did not generally favor promoters or gene body regions (Fig. 3, Supplementary Table 8). Only embryonal tumors displayed slightly varying associations. While many of the dhmCpGs associated with embryonal tumors followed similar patterns of decreased 5hmC levels and increased gene expression, there were some CpGs with decreased 5hmC and decreased gene expression, as well as CpGs with increased

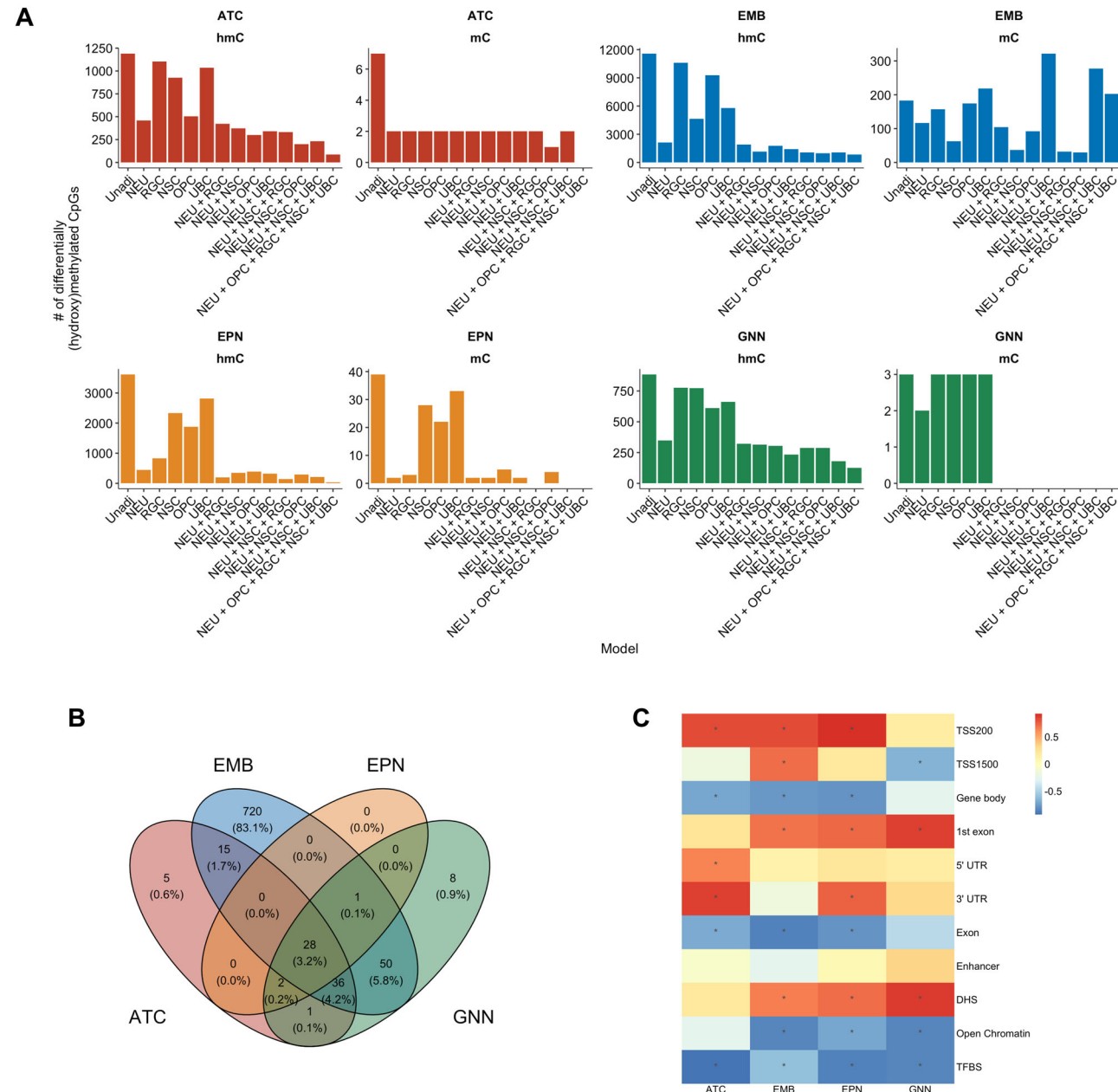

**Fig. 2 | Adjusting for proportions of cell types of interest reduces the number of differentially hydroxymethylated and methylated CpGs across tumor types compared to non-tumor pediatric brain tissue. A** The number of differentially hydroxymethylated (hmC) and methylated (mC) CpGs under $q < 0.05$ threshold in astrocytoma (ATC), embryonal tumors (EMB), ependymoma (EPN), and glioneuronal/neuronal tumors (GNN) compared to non-tumor pediatric brain tissue. X-axes indicate each cell type proportion included in the model. Each model, even 'unadjusted' models, includes sex and age at diagnosis in the linear model. **B** Venn diagram of the differentially hydroxymethylated CpGs among the different tumor types. **C** Heatmap of correlation between number of cell types included in model and proportion of dhmCpGs per genomic context. Correlation calculated by Spearman rank test. Heatmap cells with * indicate statistically significant correlation at $p < 0.05$. Source data are provided as a Source Data file.

**Table 2 | Summary of dmCpGs in unadjusted and five-cell type adjusted EWAS model**

|  | Unadjusted model dmCpGs *N* (%) | dmCpGs that are also dhmCpGs *N* (%) | Adjusted model dmCpGs *N* (%) | dmCpGs that are also dhmCpGs *N* (%) |
|---|---|---|---|---|
| Astrocytoma (ATC) | 7 (0.001%) | 3 (42.9%) | 0 (0%) | 0 (0%) |
| Embryonal (EMB) | 183 (0.04%) | 90 (49.1%) | 202 (0.04%) | 15 (7.4%) |
| Ependymoma (EPN) | 39 (0.008%) | 25 (64.1%) | 0 (0%) | 0 (0%) |
| Glioneuronal/neuronal (GNN) | 3 (0.0006%) | 1 (33.3%) | 0 (0%) | 0 (0%) |

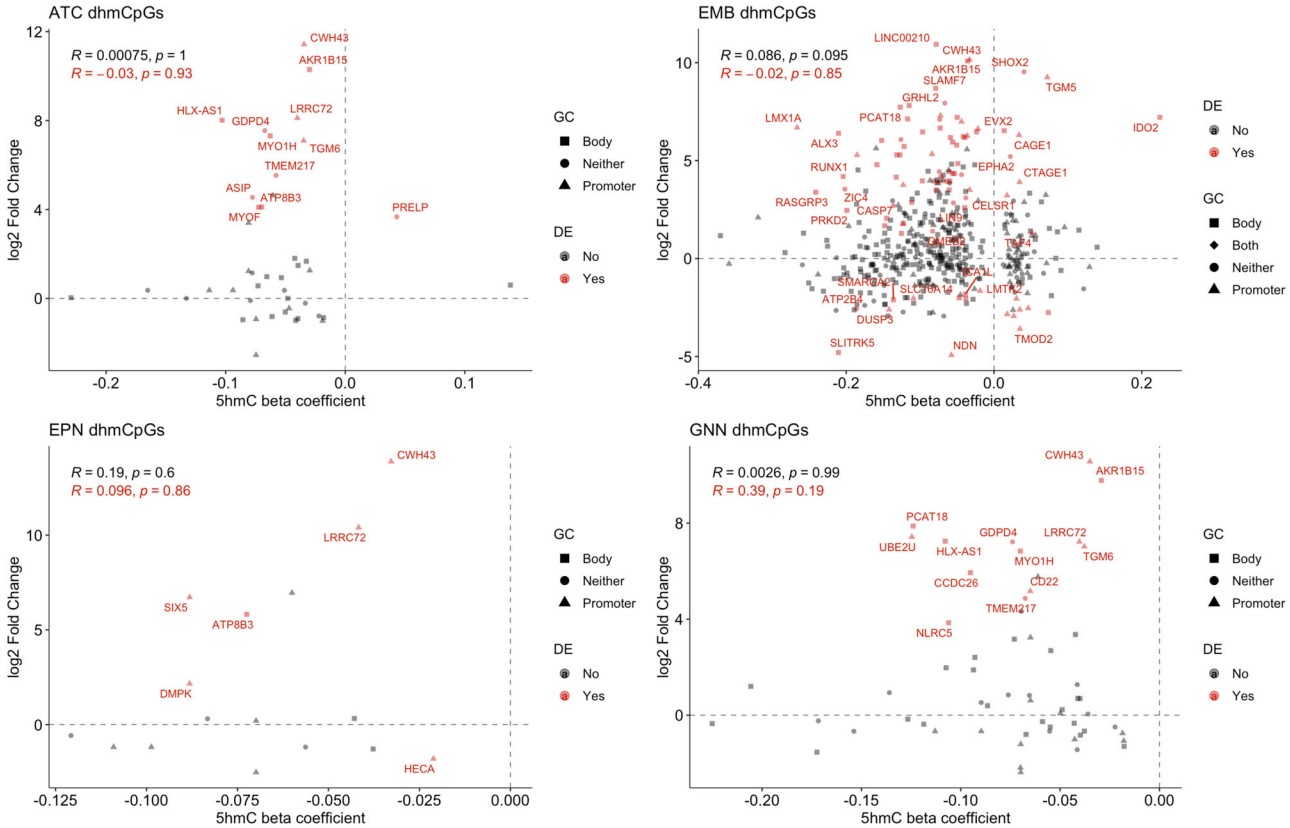

**Fig. 3 | Hypo-hydroxymethylation of CpGs is associated with changes in gene expression.** Association between differentially hydroxymethylated CpG beta coefficients and log2 fold changes in gene expression for astrocytoma, embryonal tumors, ependymoma, and glioneuronal/neuronal tumors. Red points indicate significantly differentially expressed genes. Shapes indicate genomic context of CpGs. Correlations were calculated using the Pearson method. Differential hydroxymethylated CpGs were identified from linear regression model and significant dhmCpGs were identified using $q < 0.05$ significance threshold. Log2 fold changes in gene expression were identified from negative binomial regression model and significantly differentially expressed genes were identified using adjusted $p$ value threshold <0.05. Source data are provided as a Source Data file.

5hmC with increased or decreased gene expression levels. Embryonal tumor associated dhmCpGs with significantly increased gene expression were less likely to be in promoter regions compared to dhmCpGs with significantly decreased gene expression (OR = 0.23, 95% CI = 0.064–0.78, $p = 0.01$). On the contrary, embryonal tumor associated dhmCpGs with significant increased expression were marginally more likely to be in gene body regions (OR = 2.81, 95% CI = 0.84–10.34, $p = 0.06$). We could not test for associations between promoter or gene body regions for other tumor types due to the limited number of dhmCpGs.

Interestingly, there were two CpGs with decreased 5hmC levels and increased gene expression in astrocytoma, ependymoma, and glioneuronal/neuronal tumors: cg18280362 located in the promoter region of *CWH43* and cg08278401 located in the promoter region of *LRRC72*. In addition, we investigated the association between changes in 5mC methylation and gene expression in the embryonal tumors where there were 24 dmCpGs associated with significant changes in gene expression (Supplementary Fig. 16). Unlike dhmCpGs, magnitude of changes in 5mC levels were negatively associated with magnitude of changes in gene expression for genes that did not have statistically significant gene expression changes ($R = -0.45$, $p = 0.029$) and genes with statistically significant gene expression changes ($R = -0.41$, $p = 0.0002$, Supplementary Fig. 16). While we could not conduct statistical tests to test for an enrichment of promoter/gene body regions for shared dhmCpGs with increased gene expression, there were 18 dhmCpGs with increased gene expression in non-promoter regions and 3 dhmCpGs with increased gene expression in promoter regions. Moreover, there were 9 dhmCpGs with increased gene expression not

in gene body regions and 12 dhmCpGs in gene body regions. Our results suggest potential roles of hydroxymethylation in regulating gene expression of certain pediatric CNS tumor-associated genes, that require further investigation to validate.

## Molecular alterations in pediatric CNS tumors occur in a cell type-specific and tumor type-specific manner
One of the major questions that remains unanswered in many epigenome-wide association studies is whether altered cytosine modification can be ascribed to a specific cell type. With data from single nuclei RNA-seq for these pediatric CNS tumors and non-tumor pediatric brain tissues, we sought to identify epigenomic alterations at a cell type-specific level. To reduce the number of covariates in our analysis we focused on neuronal-like and progenitor-like cell types (Supplementary Table 9). The progenitor-like cells were an aggregation of NSC, RGC, OPC, and UBC. We used an approach developed by ref. [103] called CellDMC to identify cell-type-specific differentially hydroxymethylated and methylated CpGs. We compared the epigenome of each tumor type to non-tumor tissue and used CellDMC to identify which cell type was driving the change in 5hmC and 5mC in the tumors compared to the non-tumor tissue. Overall, we identified abundant dhmCpGs for each cell type and tumor type, far greater than the scope of CpGs identified with bulk tissue EWAS (Fig. 4A, Supplementary Figs. 17–20, Supplementary Table 10, Supplementary Data 17–20). While there were a relatively lower number of dmCpGs compared to the dhmCpGs, there were some dmCpGs detected in the cell type-specific model (Fig. 4B). Majority of the cell type-specific dhmCpGs were tumor-type-specific (Fig. 4C, D, Supplementary Fig. 21).

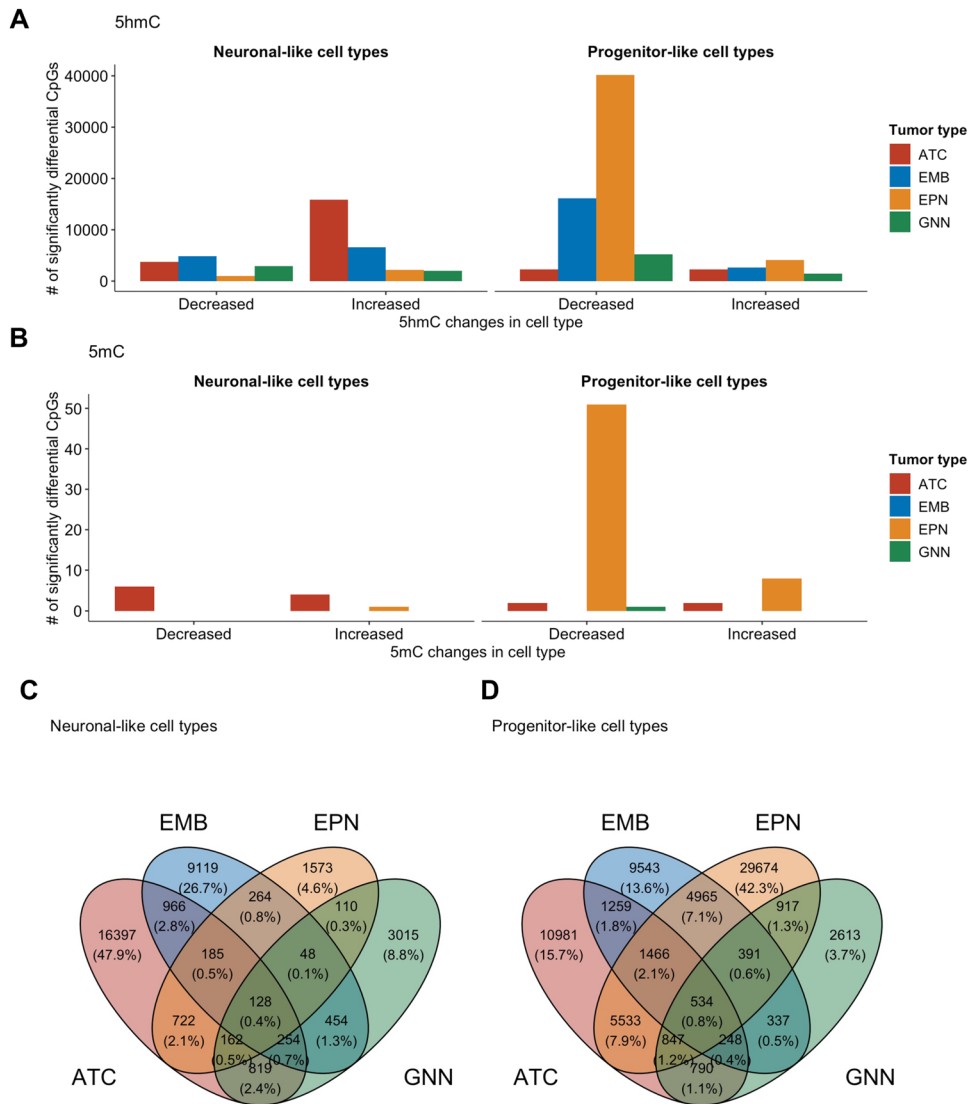

**Fig. 4 | 5hmC is altered in cell type-specific and tumor type-specific manner.** Cell type driven differentially (**A**) hydroxymethylated and (**B**) methylated CpGs in each tumor type identified by epigenome wide association study adjusted for cell type proportions. Significantly differentially hydroxymethylated and methylated CpGs were defined by *q* < 0.05. Venn diagram of shared differentially hydroxymethylated CpGs in (**C**) neuronal-like cell types and (**D**) progenitor-like cell types across the four tumor types. Number of cell type driven differential CpGs were statistically significant under adjusted *p* < 0.05. ATC Astrocytoma, EMB Embryonal tumors, EPN Ependymoma, GNN Glioneuronal/neuronal tumors. Source data are provided as a Source Data file.

However, 128 dhmCpGs were observed in the neuronal-like cell types and 534 dhmCpGs were observed to be driven by the progenitor-like cell types across all four tumor types. While some neuronal-like cell-specific driven dhmCpGs were acting on the same genes as the progenitor-like cell-specific dhmCpGs, genes that had decreased 5hmC in the progenitor-like cells were exclusive (Supplementary Fig. 22).

We then assessed the genomic context of cell type-specific dhmCpGs and tested for enrichment to various genomic contexts stratified by the direction of differential hydroxymethylation. Interestingly, both increased and decreased dhmCpGs in neuronal-like and progenitor-like cell types of astrocytoma and glioneuronal/neuronal tumors were enriched in similar contexts at DHS, 1st exons, promoter regions (TSS200, TSS1500), and 5′ UTR regions (Fig. 5, Supplementary Data 21). dhmCpGs in ependymoma were dependent on the cell type in which it was occurring. Ependymoma-associated dhmCpGs in the NEU and CpGs with increased 5hmC in progenitor-like cells were enriched in similar regions as the astrocytoma and glioneuronal/neuronal tumors. On the contrary, ependymoma-associated CpGs with decreased 5hmC in the progenitor-like cells were enriched in

transcription factor binding sites (TFBS), 3′ UTR, gene body, and exon regions. The dhmCpGs, especially for those occurring in the progenitor-like cell types, in embryonal tumors were enriched in distinct genomic contexts compared to the other tumor types. Progenitor-like cell type-specific dhmCpGs were enriched in the transcription factor binding sites, 3′ UTR, gene body, exons, and enhancers.

Our findings indicate that hydroxymethylation alterations are driven by different cell types in different tumor types.

## Cell type-specific gene expression changes associated with changes in hydroxymethylation

We next evaluated cell-specific gene expression changes for genes with cell-type-specific changes in hydroxymethylation. We calculated gene expression scores for genes associated with CpGs with differentially hydroxymethylated CpGs in the neuronal-like cells and progenitor-like cells for each granular cell types incorporated in our analysis for each tumor type (Supplementary Figs. 23–26). Interestingly, for all tumor types, the expression scores for genes associated with CpGs with

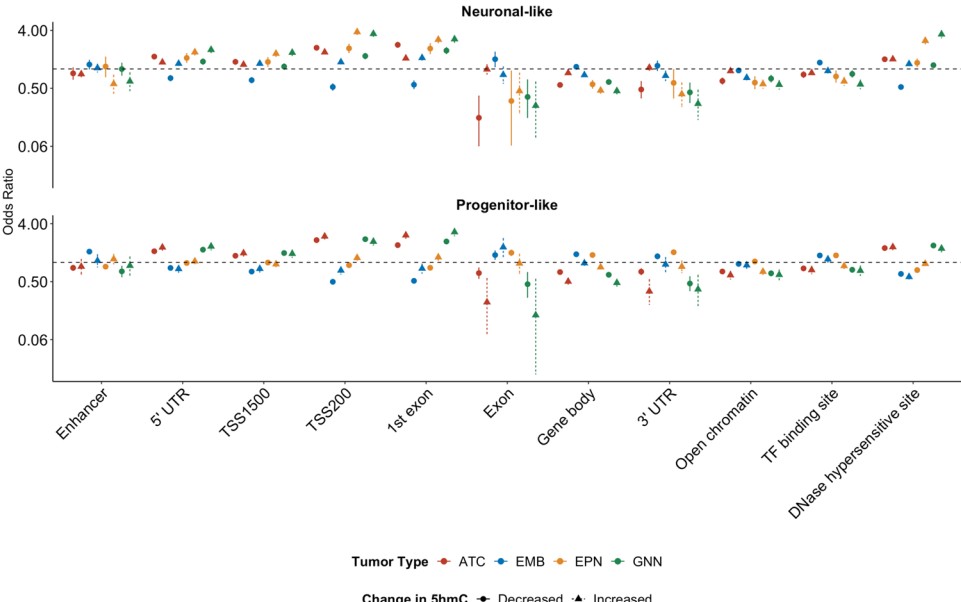

**Fig. 5 | Cell type-specific differential hydroxymethylation tumor type-specific.**
Enrichment of differentially hydroxymethylated CpGs at specific genomic contexts by tumor type and direction of differential methylation as represented by odds ratios and 95% confidence intervals. Odds ratio points and confidence intervals colored by tumor type. The direction of 5hmC change is indicated by the shape indicating the odds ratio. Odds ratios were calculated by Mantel-Haenszel test. Source data are provided as a Source Data file.

increased or decreased hydroxymethylation were increased in the OPCs of the tumors compared to non-tumor pediatric brain tissue (Fig. 6A). Only the OPCs in embryonal tumors did not show a statistically significant increase in the expression of genes with increased 5hmC in the progenitor-like cells. On the contrary, gene expression levels for each of the gene sets with cell type-specific alterations in 5hmC were decreased in each of the cell types for all tumors compared to the non-tumor pediatric brain tissue.

*HDAC4*, established as associated with cancer progression and poor prognosis in a variety of tumor types[113–121], was one gene with cell type-specific dhmCpGs across all four tumor types. Interestingly, the majority of the CpGs with decreased 5hmC were associated with progenitor-like cell types, while the majority of the CpGs with increased 5hmC were associated with the neuronal-like cell types in the tumor tissue (Fig. 6B). More than 50% of the dhmCpGs in *HDAC4* for each tumor type were in the gene body (Table 3). There were few dhmCpGs in the 5′ UTR, TSS200, and DHS. The neuronal-like cell types had lower expression of *HDAC4* across all tumor types compared to the non-tumor tissue (Fig. 6D). On the contrary, the progenitor-like cell types had higher levels of *HDAC4* expression. However, these differences in gene expression in each cell type of each tumor type compared to the same cell types in non-tumor tissues were not statistically significant which was likely due limitations from sample size (Fig. 6D).

*IGF1R* had dhmCpGs across all tumor types and is associated with tumorigenesis, therapy resistance, and poor survival in different cancer types, including in some pediatric CNS tumor types[122–132]. Most of the dhmCpGs with decreased 5hmC were associated with the progenitor-like cell types in the tumor tissue while only a couple dhmCpGs were in the neuronal-like cell types of the tumor tissue (Fig. 6C). Like *HDAC4*, the dhmCpGs in *IGF1R* were mostly located in the gene body and DHS, with a few scattered in the enhancer and 3′ UTR regions (Table 4). Consistent with the lack of changes in hydroxymethylation in the neuronal-like cell types of the tumors, gene expression levels of *IGF1R* did not differ between tumors and the non-tumor tissue among neuronal-like cell types (Fig. 6D). However, following the decreases in hydroxymethylation, *IGF1R* gene expression levels were higher in the progenitor-like cell types, particularly the

OPCs, in the tumors than in the progenitor-like cell types of non-tumor tissue. As with *HDAC4*, the differences between each cell type of each tumor type and same cell type of non-tumor tissues were also not statistically significant (Fig. 6D). EWAS results from bulk tumor tissue identified only one or two CpGs in *HDAC4* and *IGF1R* as differentially hydroxymethylated in either cell type-adjusted or unadjusted model (Table 4).

Our results suggest potential roles of hydroxymethylation of CpGs located within the gene body regions in affecting the gene expression of critical cancer genes, like *HDAC4* and *IGF1R*. However as statistical significance levels were not reached in cell type specific differences in gene expression levels likely due to limited sample size, further experimentation is needed to validate these results.

## Discussion

In this study, we investigated the cell type-specific cytosine modification alterations in pediatric central nervous system tumors with a multi-omic approach. We described the cell type composition effects that occur in epigenome-wide association studies using bulk pediatric central nervous system tumors and non-tumor pediatric brain tissue. We identified that there were more differentially hydroxymethylated CpGs associated with each tumor type, particularly in the progenitor-like cell types, rather than differentially methylated CpGs. Lastly, we show that the cell type-specific changes in hydroxymethylation are associated with cell type-specific gene expression changes in pediatric central nervous system tumors.

Based on methods to classify tumor subtypes and the predominant focus on DNA methylation, it was unexpected that there were very few differentially methylated CpGs associated with each tumor type. One possible explanation for this phenomenon may be that as these are pediatric tissues, there is still ongoing development with which 5hmC is associated. As our results suggest the epigenome-wide alterations of 5hmC in these tumors, it may be critical to distinguish between 5mC and 5hmC to better understand the molecular underpinnings of these pediatric CNS tumors. Furthermore, it may be beneficial to incorporate 5hmC into cytosine modification-based classification methods to improve performance.

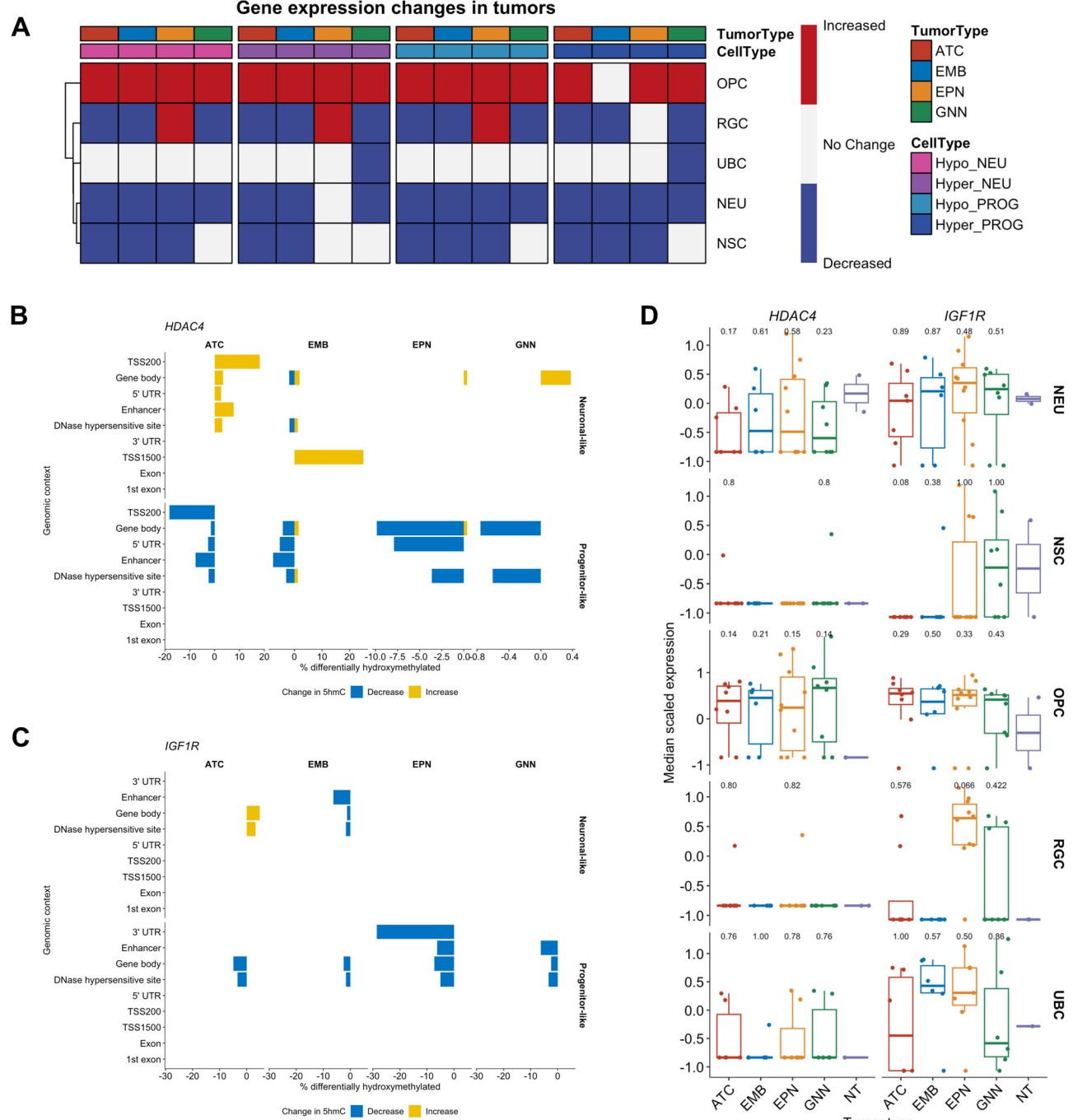

**Fig. 6 | Alterations in hydroxymethylation are associated with cell type-specific changes in gene expression. A** Summary heatmap of changes in gene expression in the gene sets with differentially hydroxymethylated CpGs per cell type. The proportion of differentially hydroxymethylated CpGs associated with (**B**) *HDAC4* and (**C**) *IGF1R* at each genomic context across the different tumor types in neuronal-like cell types and progenitor-like cell types. Blue bars indicate the proportion of hydroxymethylated CpGs that are decreased in the tumors. Yellow bars indicate the proportion of hydroxymethylated CpGs that are increased in the tumors. **D** Gene expression levels of *HDAC4* and *IGF1R* for each cell type across the tumor types and non-tumor tissue. Sample size: ATC = 8, EMB = 6, EPN = 10, GNN = 8. Differences between each tumor type to non-tumor tissue were determined by Wilcoxon rank sum test. Number above the boxplot indicates *p* value from the Wilcoxon rank sum test. In the boxplots of (**D**), the low ends of the segment indicate the minimum and the high ends of the segment indicate the maximum. Lower bounds of the box indicate the 25th percentile and the higher bounds of the box indicate the 75th percentile. Segment in the middle is the median. Source data are provided as a Source Data file.

Pediatric tumors are known not to have substantial genetic alterations[133]. Our results suggest that pediatric CNS tumors may be characterized by non-mutational epigenomic reprogramming[134,135]. We identified a substantial number of differentially hydroxymethylated CpGs associated with progenitor-like cell types of each tumor type. Additionally, even among the shared differentially hydroxymethylated CpGs in the progenitor-like cell types, numerous differentially hydroxymethylated CpGs were located within different genes that regulate epigenetic patterns, such as *DNMT3A, HDAC4, MLLT3,* and *KAT2B*. Furthermore, pediatric brain cancers have been shown to contain somatic mutations in epigenetic regulator genes such as *H3F3A, KDM6A,* and *MLL3*[136–138]. Considering the dysregulation of the

**Table 3 | Genomic context of dhmCpGs in (A) *HDAC4* and (B) *IGF1R* for each tumor type**

**A) HDAC4**

|  | TSS200 | TSS1500 | Gene body | 1st exon | 5' UTR | 3' UTR | Exon bound | Enhancer | DHS | dhmCpG total |
|---|---|---|---|---|---|---|---|---|---|---|
| ATC | 2 (15%) | 0 (0%) | 10 (77%) | 0 (0%) | 1 (8%) | 0 (0%) | 0 (0%) | 1 (8%) | 5 (38%) | 13 |
| EMB | 0 (0%) | 1 (5%) | 16 (84%) | 0 (0%) | 2 (11%) | 0 (0%) | 0 (0%) | 1 (5%) | 9 (47%) | 19 |
| EPN | 0 (0%) | 0 (0%) | 27 (90%) | 0 (0%) | 3 (10%) | 0 (0%) | 0 (0%) | 0 (0%) | 6 (20%) | 30 |
| GNN | 0 (0%) | 0 (0%) | 2 (100%) | 0 (0%) | 0 (0%) | 0 (0%) | 0 (0%) | 0 (0%) | 1 (50%) | 2 |

**B) IGF1R**

|  | TSS200 | TSS1500 | Gene body | 1st exon | 5' UTR | 3' UTR | Exon bound | Enhancer | DHS | dhmCpG total |
|---|---|---|---|---|---|---|---|---|---|---|
| ATC | 0 (0%) | 0 (0%) | 4 (100%) | 0 (0%) | 0 (0%) | 0 (0%) | 0 (0%) | 0 (0%) | 2 (50%) | 4 |
| EMB | 0 (0%) | 0 (0%) | 3 (100%) | 0 (0%) | 0 (0%) | 0 (0%) | 0 (0%) | 1 (33%) | 2 (67%) | 3 |
| EPN | 0 (0%) | 0 (0%) | 6 (75%) | 0 (0%) | 0 (0%) | 2 (25%) | 0 (0%) | 1 (13%) | 3 (38%) | 8 |
| GNN | 0 (0%) | 0 (0%) | 2 (100%) | 0 (0%) | 0 (0%) | 0 (0%) | 0 (0%) | 1 (50%) | 2 (100%) | 2 |

**Table 4 | Comparison of the number of differentially hydroxymethylated CpGs in *HDAC4* and *IGF1R* identified by bulk tissue EWAS and CellDMC for each tumor type**

|  | Tumor type | Bulk EWAS (CT unadjusted) | Bulk EWAS (CT adjusted) | CellDMC (Neuronal-like) | CellDMC (Progenitor-like) |
|---|---|---|---|---|---|
| *HDAC4* | ATC | 0 | 0 | 12 | 7 |
|  | EMB | 1 | 1 | 11 | 17 |
|  | EPN | 1 | 0 | 1 | 30 |
|  | GNN | 0 | 0 | 1 | 2 |
| *IGF1R* | ATC | 0 | 0 | 4 | 4 |
|  | EMB | 2 | 0 | 1 | 2 |
|  | EPN | 1 | 0 | 0 | 8 |
|  | GNN | 0 | 0 | 0 | 2 |

epigenome may be important when developing innovative therapeutic strategies for these tumors.

While much more investigation has been conducted into how DNA methylation regulates gene expression, less is known about how DNA hydroxymethylation can also be associated with changes in gene expression. We identified relationships between cell type-specific hydroxymethylation patterns and cell type-specific gene expression in our pediatric CNS tumors. Our findings indicate that hydroxymethylation changes in the gene body regions can alter gene expression. Previous studies have found positive associations between DNA methylation in gene body regions and gene expression changes[33,44]. However, many genome-wide DNA methylation studies use the traditional bisulfite treatment approach to measure 5mC. Because bisulfite treatment alone cannot distinguish between 5mC and 5hmC[25], some methylation signals may have been from 5hmC. Further studies that explicitly distinguish between 5hmC and 5mC are needed to gain a clearer understanding of the effects of DNA cytosine modifications on gene expression.

We identified two genes, *HDAC4* and *IGF1R*, in our pediatric CNS tumors that were both epigenetically and transcriptionally altered in comparison to non-tumor pediatric brain tissue. *HDAC4* and *IGF1R* had differentially hydroxymethylated CpGs and increased expression in OPCs across all four of our tumor types. Our results suggest a potential role of hydroxymethylation regulating genes associated with tumorigenesis. With these targets already having been studied in adult cancers, there are pharmacological inhibitors that already exist for these targets. Our study expands previously suggested ideas of targeting *HDAC4* and *IGF1R* in certain pediatric CNS tumor types[127,139,140].

Accruing a large sample size for pediatric CNS tumors is particularly difficult as they are very rare in the general population. The limited sample size prevented us from including other potential variables like tumor location. As different parts of the brain may be composed of differing cell types, not adjusting for tumor location introduces limitations in our conclusions. However, as we compare the epigenome within major cell types, we believe that some limitations of not including tumor location were addressed. Furthermore, the limited sample size reduced our statistical power in our analyses. While our study does incorporate a reasonable sample size for these rare tumors, the smaller sample size limited the inclusion of other variables and cell types that may affect methylation and transcription into our models. Moreover, our study incorporates multiple genome-wide and epigenome-wide molecular features of the matched tumor sample to give a more comprehensive landscape of each tumor type. Multi-omic approaches involving single nuclei RNA-seq, bulk RNA-seq, 5mC, 5hmC epigenome profiles of different pediatric CNS tumors have been limited in investigations to our knowledge.

Future studies with an expanded cohort of pediatric CNS patients will allow us to assess the epigenomic alterations in additional cell types of interest, such as glial cells. Moreover, following our findings of cell type-specific changes in DNA cytosine modifications in these pediatric CNS tumors, other tumor types may also have cell type-specific that have yet to be detected. Tools to understand the cell type composition of tissues should be incorporated in bulk epigenome-wide association studies to discriminate the cell type composition effects.

Our study addresses gaps that currently exist in understanding epigenomic alterations at the cell type level in pediatric central nervous system tumors. Changes in hydroxymethylation were particularly drastic in progenitor-like cells and were associated with cell type level alterations in transcription. We highlight the relevance of epigenome dysregulation in pediatric central nervous system tumors that may lead us to more effective therapeutic targets.

## Methods

This study complies with all Dartmouth Hitchcock Medical Center Institutional Review Board regulations. This study was approved by the Dartmouth Hitchcock Medical Center Institutional Review Board Study #00030211. Parents/legal guardians of the subjects provided consent for the use of tissues for research purposes.

### Sample information

Cytosine modifications, bulk tissue gene expression, and single nuclei gene expression were measured in 32 pediatric CNS tumors of various types and 2 non-tumor pediatric brain tissues (Table 1, Supplementary Table 1). Only samples with all four molecular measurements were included in downstream analyses. The samples were collected from patients being treated at Dartmouth-Hitchcock Medical Center and the Dartmouth Cancer Center from 1993 to 2017. For each tumor type, the number of samples was distributed evenly with 8 samples for astrocytoma, 6 for embryonal tumors, 10 for ependymoma, and 8 for glioneuronal/neuronal tumors. Pathological re-review for the

histopathologic tumor type and grade were done according to the 2021 World Health Organization CNS tumor classification system, then categorized into broader tumor types. The non-tumor pediatric brain tissues were obtained from patients who underwent surgical resection for epilepsy.

## Data collection and pre-processing

**Single nuclei RNA-sequencing.** Nuclei were isolated from fresh frozen tissue samples following the Nuclei Pure Prep nuclei isolation kit (Sigma-Aldrich, St. Louis, MO) with some modifications The samples were first washed with PBS to remove extraneous OCT the samples were frozen in. The tissue was homogenized with both wide and narrow pestles submerged in 2.5 mL of the lysis buffer in a Dounce homogenizer. The lysate mixed with 4.5 mL 1.8 M sucrose cushion were gently layered on top of the 2.5 mL of 1.8 M sucrose cushion in Beckman ultracentrifuge tubes. Samples were centrifuged for 45 min at 22,673 g at 4 °C in an ultracentrifuge. Each sample was multiplexed with lipid-tagged oligonucleotides following the MULTI-seq protocol[141]. Libraries for single nuclei RNA-seq were prepared following the 10X Genomics Single Cell Gene Expression workflows (10X Genomics, Pleasanton, CA). Libraries were pooled and sequenced using the Illumina NextSeq500 instrument. 10X Cell Ranger software was used to align sequences to the GRCh38 pre-mRNA reference genome.

Low-quality nuclei, as defined as having greater than 10,000 and less than 2000 features and more than 5% of reads that map to mitochondrial genes, were removed for analyses. Samples were demultiplexed using an integrative approach, combining barcode based demultiplexing and genotype-based demultiplex method[142,143]. Pooled nuclei were demultiplexed by hashtag oligonucleotides using HTO-Demux function in Seurat v4[142,144–146]. Pooled samples were also demultiplexed using Vireo, a genotype based demultiplexing method[143]. We performed genetic demultiplexing analysis using genotype data following the methods described in ref. 147, implemented in a Nextflow workflow[148]. Briefly, bulk RNA-seq reads from each sample were mapped to the reference genome (GRCh38.p13) using STAR[149]. Pooled single-nuclei RNA-seq reads were mapped to the reference genome using STARsolo[150]. Variants among the samples within each pool were identified and genotyped with bcftools mpileup[151] using the mapped bulk reads. Individual cells were then genotyped only at the sites identified using the bulk RNA using cellsnp-lite (mode 1a)[152]. Cell genotypes were used to identify the sample of origin for each cell using Vireo[143]. Code for the genetic demultiplexing workflow can be found at https://github.com/AlexsLemonade/alsf-scpca/tree/main/workflows/genetic-demux.

To integrate the methods, we first used sample identity assigned from the hashtag oligonucleotides. If the nuclei were confidently assigned a sample, it was compared to the genotype-based sample assignment. Those that did not match the same sample were filtered out. If the nuclei were assigned as a doublet or to none of the samples, the nuclei were assigned to a sample based on the genotype-based approach. 84,700 nuclei with confident sample assignment were used in analysis.

Downstream analyses for single nuclei-RNA seq were done with the Seurat package v4 in R[142,144–146]. Cell types for the nuclei were assigned by expression levels for classical markers for brain cell types such as *GFAP* and *AQP4* for astrocytes and *MOG* and *PLP1* for oligodendrocytes. The cell types were then validated by using the Variance-adjusted Mahalnobis method, a gene set enrichment testing developed to be specific to singe cell RNA-seq data, with gene sets derived from specific brain cell types[153]. Further details for single cell RNA-seq pre-processing and analysis are detailed in ref. 110.

**Bulk RNA-sequencing.** Unused nuclei from our single nuclei RNA-seq experiment were used for bulk RNA-sequencing. RNA was isolated following the RNeasy Plus kit (Qiagen, Hilden, Germany). Libraries for bulk RNA-seq were prepared following the Takara Pico v3 low-input protocol (Takara Bio, Kusatsu, Japan).

Quality control for raw single-end RNA-seq data was checked using FastQC v0.11.8[154]. Reads were trimmed of polyA sequences and low-quality bases using Cutadapt v2.4[155]. Reads were aligned to the human pre-mRNA genome GRCh38 with STAR v2.7.7a[149]. Quality control of aligned reads was confirmed with *CollectRNASeqMetrics* in the Picard software v2.18.29[156]. Duplicate reads were identified with *MarkDuplicates* function in the Picard software[156]. One sample with an exceedingly high duplicate read percentage was removed from downstream analyses. Counts per gene were estimated using the *htseq-count* function in the HTSeq software v0.11.2[157].

**DNA methylation and hydroxymethylation.** In total, DNA from 32 paired pediatric brain tumor and 2 non-tumor brain samples was treated with tandem bisulfite and oxidative bisulfite conversion followed by hybridization to Infinium HumanMethylationEPIC BeadChips to measure DNA methylation (5mC) and hydroxymethylation (5hmC). Raw BeadArray data were preprocessed using the *SeSAMe* pipeline (v1) from Bioconductor, including data normalization and quality control[158]. Cross-reactive probes, SNP-related probes, sex chromosome probes, non-CpG probes, and low-quality probes (pOOBHA > 0.05) were masked in the analysis[159]. The *oxBS.MLE* function was used to infer 5mC and 5hmC levels[160].

**Tumor purity estimates.** Tumor purity for the tissue samples with DNA cytosine modifications was estimated using the *getPurity* function with the non-tumor pediatric tumor tissue as our non-tumor reference and the low-grade glioma option as our cancer type in the InfiniumPurify package v1.3.1 in R[161].

## Statistical analyses

Distribution of tumor tissues 5mC and 5hmC were compared to distribution of non-tumor 5mC and 5hmC, respectively, using a Kolmogorov-Smirnov tests. Distributions were considered to be statistically significant at $p < 0.05$ threshold. Outliers for MDI were determined using the Grubb's test for outliers at statistical significance threshold of $p < 0.05$. Linear regression models were used to determine association between 5hmC and 5mC Methylation Dysregulation Index values with genomic context and tumor type. Linear regression models were run with the *lm* function in the stats package in R.

**Epigenome-wide association studies.** Linear regression models, adjusting for sex, age at diagnosis, and tumor purity in all models, were used to identify differentially methylated and hydroxymethylated CpGs associated with each tumor type compared to the non-tumor tissue. Due to sample size, tumor location was not adjusted for in the linear regression models. Multiple linear regression models, with adjustments for different cell type proportions identified from the single nuclei RNA-seq data, were added to the models. Linear regression models were fit by using *lmFit* and *eBayes* functions in the limma (v3.54.2) package in R[162]. CpGs were considered differentially methylated or hydroxymethylated under the q-value threshold of 0.05.

Cell type-specific differential hydroxymethylation and methylation for each tumor type were identified using CellDMC[103]. CellDMC is a statistical model that identifies both differentially methylated CpGs and which cell type drives the differential methylation by incorporating cell type proportions as interaction terms in the linear regression model in the epigenome wide association study[103]. *CellDMC* was conducted within the EpiDISH (v2.14.1) R package[103]. Proportions of cell types of interest (neurons and progenitor-like cell types) were pulled from the single nuclei RNA-seq dataset. To limit overfitting the model in our relatively smaller sample size, we aggregated the progenitor-like cell types into a single cell type category. The progenitor-like cell types included NSC, RGC, OPC, and UBC. UBCs were included due to the

high levels of stemness score in the cell types identified previously. Separate models to compare each tumor type to the non-tumor tissue were run with the same cell types (progenitor-like and neuronal-like cell types) included in each model.

**Differential gene expression testing.** Negative binomial regression models were used to identify the differential expressed genes in each tumor type compared to non-tumor tissue. One model was fit adjusting for age at diagnosis and sex. The other model was fit adjusting for age at diagnosis, sex, and the proportions for cell types of interest (NEU, NSC, RGC, OPC, UBC). Negative binomial models were fit by using *DESeq* function in the DESeq2 package v1.36.0 in R[163]. Genes were considered as differentially expressed under the adjusted p-value threshold of 0.05.

**Pathways enrichment testing.** Reactome pathways enrichment associated with differentially expressed genes in each tumor type were identified using the *enrichPathway* function in the ReactomePA package v1.40.0 in R[164].

**Genomic context enrichment test.** Enrichment tests for genomic context for differentially hydroxymethylated CpGs were conducted using the Mantel-Haenszel (MH) test. The MH test was adjusted for the type of probe (Type I or Type II) used for the CpG in the Illumina Methylation EPIC array.

### Reporting summary

Further information on research design is available in the Nature Portfolio Reporting Summary linked to this article.

## Data availability

The raw single nuclei-RNA seq data and the processed data for single nuclei-RNA seq generated in this study are available in the Gene Expression Omnibus under accession code GSE211362. The raw hydroxymethylation/methylation data generated in this study have been deposited in the Gene Expression Omnibus under accession code GSE152561. The raw bulk RNA-seq data generated in this study have been deposited in the Gene Expression Omnibus under accession code GSE241396. Source data are provided as a Source Data file. All larger size source data files are available at https://figshare.com/projects/Associations_in_cell_type-specific_hydroxymethylation_and_transcriptional_alterations_of_pediatric_central_nervous_system_tumors/193781. GRCH38 reference data are available in the National Library of Medicine database (https://www.ncbi.nlm.nih.gov/datasets/genome/GCF_000001405.26/). Source data are provided with this paper.

## Code availability

Code used for analysis is available at https://github.com/sarahmklee/IntegrativePCNS.

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

## Acknowledgements

This work was supported by a Prouty Pilot award from the Dartmouth Cancer Center and a Single-cell Pediatric Cancer Atlas (ScPCA) grant from the Alex's Lemonade Stand Foundation. M.K.L. was supported by the Burroughs-Wellcome Fund: Big Data in the Life Sciences at Dartmouth. N.A. was supported by the S.M. Tenney Fellowship at Dartmouth. This work was also supported by National Institutes of Health (R01CA216265, R01CA253976, and P20GM104416 – 6369) to B.C.C. and P20 GM104416-09/8299 and CDMRP/Department of Defense (W81XWH-20-1-0778) to L.A.S. Single nuclei RNA-seq experiments were conducted in the Genomics and Molecular Biology Shared Resource (GMBSR) at Dartmouth, which is supported by NCI Cancer Center Support Grant 5P30CA023108 and NIH S10 (1S10OD030242) awards. Single-nuclei RNA experiments were also supported through the Dartmouth Center for Quantitative in collaboration with the GMBSR with support from NIGMS (P20GM130454) and NIH S10 (S10OD025235) awards.

## Author contributions

M.K.L., N.A., and B.C.C. designed the study. N.A., G.J.Z., and L.N. identified subject populations and collected tissue samples. M.K.L., N.A., and L.P. performed experiments to collect cytosine modification and gene expression data. M.K.L., N.A., and F.W.K. performed experiments to collect single nuclei-RNA seq data. M.K.L. and Z.Z. processed data for downstream analyses. M.K.L. performed statistical analyses under the supervision of L.A.S. and B.C.C. B.C.C. supervised the project. All authors reviewed the manuscript.

## Competing interests

B.C.C. is an advisor to Guardant Health which had no role in this work. All other authors declare no competing interests.
