## [Peer Review File · Nature Communications]

REVIEWER COMMENTS

Reviewer #1 (Remarks to the Author): Expert in paediatric CNS tumour genomics, epigenomics, and methylation

In this study, Drs. Lee et al examined hydroxymethylation alterations in a set of pediatric CNS tumors as associated with cell-type specific transcriptional changes. The biological problem that they attempted to address is important for childhood brain cancers and can potentially deepen our understanding on intra-tumoral cellular heterogeneity (cell type composition). The strategy of integrated analysis of hydroxymethylation and single nuclei-RNAseq represent a new application in pediatric brain tumors. The bioinformatic approaches were well designed and nicely executed. Their discovery of a preponderance differential CpG hydroxymethylation rather than methylation is interesting. There are, however, some concern about this study:

Major concerns:

1. One of the major concerns is the data presentation and manuscript preparation. Many of the key information was missing in the text of the result section, as there were no tumor types that were mentioned and no description of the number of normal and tumor tissues involved, making it very difficult to evaluate their data.
2. A second major concern is the very limited tumor samples. Although it is well known that pediatric brain tumor tissues are very difficult to obtain, it is also very well established that these tumors are composed of many molecularly distinct subtypes. With a small collection of 32 tumor samples that included tumors of different locations, different pathological diagnosis, different grade (high and low), there is a lack of statistic power to draw any meaningful conclusions from this study.

Minor concerns:

1. In Figure 2E, were the shared dhmcpgs (90 and 10/4%) also detected in normal brain tissues? What were differences among tumors of different locations?
2. Were the normal tissues age matched and location matched?
3. On page 9, 1st paragraph, "Our results indicate that hydroxymethylation may be associated with changes in gene expression for certain genes in pediatric CNS tumors" does not mean much. It is hard to believe any conclusion can be drawn based on such loose correlations.
4. This is the first time to see someone using ACT as an abbreviation for astrocytoma.
5. This study emphasizes the "cell type-specific" cytosine modification, but the "progenitor-like cells were an aggregation of neural stem cells, radial glial cells, oligodendrocyte precursor cells and unipolar brush cells", not strictly cell-type specific.

Reviewer #3 (Remarks to the Author): Expert in single-cell multi-omics and brain cancer genomics

In this manuscript, Kyung Lee et al. aimed to evaluate cell-type alterations in methylation (5mc and 5hmC) in a variety of pediatric tumour tissue relative to non-tumor tissue, as well as their association with gene expression level.

The findings related to 5-hmC are interesting. However, with bulk methylation data and limited CpG sites captured, the authors should be cautious in interpreting the results. Further, even though there are some novel findings from a biological standpoint, some of the statements and results are not adequately supported by quantitative/statistical measurements. Several of the findings must be interpreted with greater precision.

Overall, the paper would benefit greatly from a more detailed explanation of the rationale and scope of the analyses, a stronger and more detailed description of the methodology, and the addition of specifics to the legends and figure panels.

The following points should be clarified and edited in the study:

1. Have the authors considered comparing their results to other single-cell sequencing studies on pediatric CNS tumors? This is a small cohort, so it would be very important that this small cohort is representative of larger patient cohorts but not a collection of extreme cases.
2. Could the authors provide more details on the implementation of CellDMC?
3. The authors should consider validating the results by sorting the cell type and measure the methylation, or using single cell methylation assays?
4. Is this the first study used different methods to identify cell type-specific alterations to cytosine modifications in pediatric CNS tumors? If not, discussion should incorporate the result from some previous research. In Results, it was briefly mentioned that “it was quite unexpected that the observed differences were almost solely in hydroxymethylation and not in methylation”.

5. Only two non-tumor tissue were included in the study, which is rather few. Including at least another non-tumor tissue would strengthen the conclusions presented in the manuscript.

6. The Methods would benefit from a more detailed description. The provided information is not enough to reproduce the results reported in this manuscript.

7. The analytical codes used to generate the main results of the paper should be provided.

8. It would be beneficial to include details on number and type of pediatric CNS tumor samples upfront, for instance on page 5, when first introducing the results. This would give the reader a better understanding on the cohort of sample types available for analyses upfront.

Along this line, reference to Supplementary Table 1 is missing from the main text. Further, number of samples reported in Table 1 differs compared to Supplementary Table 1 (see number of Astrocytoma samples). In the Methods section, it is mentioned that 32 samples were measured, but 33 samples were treated with DNA methylation and hydromethylation assays.

9. Figure 1A is not very intuitive (median beta values of 5-mC have higher proportions around 0 - 0.1 and 0.8 - 1, while median beta values of 5-hmC fall into 0 - 0.2?).

What is the statistical significance of tumor vs. non-tumor tissue?

May also consider switch the order of the legend? (i.e., 5-mC tumor & non-tumor, 5hmC tumor & non-tumor).

10. In figure 1B and D, a visualization that includes inter-quartile ranges (e.g., boxplots) would make the assessment of outlier samples easier to detect and identify for the reader. How did the authors identify outliers to be removed in the analysis of Supplemental Figure 2?

11. In figure 1C and Supplementary Figure 2C, it would be beneficial to color code the dots by tumor type for easier assessment of the results. Also, $R = 0.44$ is a little bit concerning. Most of the data points for 5-hmC MDI reside around 0.04 while 5mC MDI from 0.04 to 0.12.

12. The authors should better contextualize the results in Figure 1A. In cancer, generally, it is common to observe globally decreased methylation in tumor tissue compared with normal, with a focally increased

methylation of promoters or CpG islands. It would be interesting to repeat analyses of Figure 1 to also report 5-mC, 5-hmC and MDI scores at specific genomic regions to compare to epigenome-wide scores.

13. The identification of the cell type composition of pediatric CNS tumor tissue and non-tumor pediatric brain tissue is an important component of the analysis; however, methods on how the cell types are identified using single-cell nuclei RNA sequencing is lacking. In general, a more in-depth description of the Methods would help the work to be reproduced. Are the cell types common across all these pediatric CNS tumor tissues? Or each pediatric tumor type is expected to have its own cell type labels?

14. "Based on the cell type proportion distributions for all of our samples, we identified neuronal-like cells (NEU), neural stem cells (NSC), oligodendrocyte precursor cells (OPC), radial glial cells (RGC), and unipolar brush cells (UBC) as having the most variance".

It is unclear what is the scope and rationale behind the analysis of Supplementary Figure 4. From Supplementary Figure 4A, it is also unclear how the authors can state that UBC and NSC have the most variance when they are centered around 0 on PC1 and PC2 axes. Is this pseudo-bulk with one data point representing one cell type from snRNA-seq?

From Supplementary 4A, it shows only OPC, NEU, and RGC as having the most variance; UBC and NSC are clustering with the rest of cell types. Again, in general, the method and approach implemented to obtain this specific result are unclear.

It would be interesting to compare the "degree of heterogeneity" between the single-cell sequencing datasets as a more appropriate measure of "variance". Ecological measures like Shannon diversity would work great in assessing whether a given tumor type is more heterogeneous vs. others.

15. Supplementary Figure 4 legends is not reflecting the panels. Missing panel B or C description in the legends and Supplementary Figure 4C is not referenced in the main text.

Supplementary 4B looks like clustering of cell types (with two PCs in the axis). The legends sound like proportions or differences? Not very clear what is done from S4A to S4B. (It is likely that they are referring to S4C as S4B currently.)

16. "For each tumor type we compared proportions of cell types with non-tumor pediatric brain tissue. Supporting our principal component analysis, the cell types with the greatest differences were NEU, NSC, OPC, RGC, and UBC".

From supplementary Figure 4C, most of the p-values are not significant (likely due to lack of power due to limited sample size). The relevance of this result is unclear. The cell types highlighted by the authors seem to have a strong tumor-type-specific enrichment. NEU is mainly present only in NT. NSC is mainly present (at low levels) in EMB. OPC is mainly present in ATC. RGC is mainly present in EPN. UBC is only present in EMB. The authors should better clarify the methods, approaches, rationale, scope and better describe the relevance and meaning of the results obtained.

17. "When all five cell types (NEU, NSC, OPC, RGC, and UBC) were incorporated into the model, we observed low number of dmCpGs associated with each tumor type."

Even when the model is not adjusted, lower number of dmCpGs is observed?

If such minor difference is observed between the 5mc status, does that suggest the previously reported dmCpGs is likely only associated with 5hmc?

Then, this is contradicting with the claim in the introduction "it has been indicated only 5-methylcytosine signal from oxidative bisulfite-treated DNA alters the classification". Please clarify.

18. In the analysis of Figure 2, how is the cell-type specificity (see Supplementary Figure 4C) considered? The number of differentially hydroxy(methylated) CpGs would likely differ as a function of a given cell type? How is the different number of samples considered in the analysis (e.g., only two non-tumor brain tissue)?

Could the authors compute a differential analysis of a given cell type between tumor and non-tumor tissue to overcome the cell type composition confounding factor?

It would be interesting to repeat this analysis by focusing on a specific genomic region, e.g., promoters or CpG islands to assess whether these results are different compared to adopting a genome-wide approach.

Can the authors provide more details on the 28 dhmcPcGs that were found in common across all samples?

Overall, as for Figure 1, the rationale of the analyses reported in Figure 2 are not well laid out. What is the mere number of differentially CpGs tell us in terms of advancing biological knowledge of pediatric tumor tissue vs. non-tumor tissue?

19. For the analysis of bulk RNA-seq differential gene expression, the authors report that the majority of differentially expressed genes were increased in expression compared to the non-tumor pediatric brain tissue. better specify the cutoff: $-\log_{10}(p\text{-value}) = 2$? Did not find any information in the figure legend.

Also, could this reflect the imbalanced dataset, given that only 2 non-tumor tissue are available? Have the authors tried to repeat the analysis by equalizing sample size to see whether this finding holds true?

What do the findings of “keratinization” vs. “insulin receptor signaling” mean biologically and in the context of pediatric tumor tissue?

20. It is unclear how the authors selected the genes to be reported in Figure 3. As already mentioned for Figure 1 and Figure 2, the lack of detail in describing the analysis and methods make it difficult to assess and interpret the robustness of the results.

21. A more quantitative approach to support the statements related to Figure 3 is needed. The authors should quantify whether there is indeed a preferential enrichment of genes that are both highly expressed and show negative 5hmC beta coefficient. Many of the statements related to Figure 3 are vague and lack direct quantification using the methylation and expression data available.

22. It is unclear how the cell-type specific differential CpGs are found in the analysis of Figure 4. What are the group comparisons to determine what is decreased vs increased? Are the authors comparing the same cell type (e.g., neuronal-like) between tumor and non-tumor tissue? Once again, more detail in the Methods, main text and Legends would greatly help the reader in understanding the results and findings.

23. The analysis of Figure 4 seems to be confounded or mainly driven by the number of cells/nuclei observed in each tumor type. For instance, it is not surprising to see that the >50% of cell-type differential CpGs for the progenitor-like cell types are from EMB and EPN, being the two tumors with the highest number of progenitor-like cells as reported in Supplementary Table 4.

This same concern applies to Supplementary Figure 20 given the big imbalance between number of neuronal-like vs. progenitor-like cell types. What is the p-value of the OR reported in Figure 5?

Overall, the data presented should be revised to support the claim that “most of the hydroxymethylation alterations occur in the progenitor-like cell types and are tumor-type-specific”.

24. It would be good to use a score or statistics in Figure 6B to show the enrichment.

25. Lack of statistical testing in Figure 6D makes it difficult to support the claim that the progenitor-like cell types had higher levels of HDAC4 expression and that IGF1R gene expression levels were higher in the progenitor-like cell types.

All in all, the results reported as is in Figure 6 do not strongly support the claim that “Our results suggest potentially critical roles of hydroxymethylation of CpGs located within the gene body regions in regulating the gene expression of critical cancer genes, like HDAC4 and IGF1R”.

In Figure 6D, the authors reported expression for “other glioma”. The samples in this category need to be specified.

26. “Our results suggest that pediatric CNS tumors may be characterized by non-mutational epigenomic reprogramming more so than genomic aberrations”. None of the analyses reported in this manuscript included genomic aberrations so this statement should be toned down or omitted.

27. “Pediatric brain cancers have been shown to contain somatic mutations in epigenetic regulator genes such as H3F3A, KDM6A, and MLL3”. Are any other data type available from these patient samples? If genomic information is available, the authors could/should investigate it to correlate genomic aberrations with the observed epigenetic aberrations.

28. The authors should provide additional supplementary tables related to the differential CpGs analysis and differential gene expression (e.g., the mentioned 702 genes, etc...).

29. Overall, figures and figure legends lack detail and specificity (e.g., how many samples? How many cells? how many genes?).

Reviewer #1 (Remarks to the Author): Expert in paediatric CNS tumour genomics, epigenomics, and methylation

In this study, Drs. Lee et al examined hydroxymethylation alterations in a set of pediatric CNS tumors as associated with cell-type specific transcriptional changes. The biological problem that they attempted to address is important for childhood brain cancers and can potentially deepen our understanding on intra-tumoral cellular heterogeneity (cell type composition). The strategy of integrated analysis of hydroxymethylation and single nuclei-RNAseq represent a new application in pediatric brain tumors. The bioinformatic approaches were well designed and nicely executed. Their discovery of a preponderance differential CpG hydroxymethylation rather than methylation is interesting. There are, however, some concern about this study:

Major concerns:

1. One of the major concerns is the data presentation and manuscript preparation. Many of the key information was missing in the text of the result section, as there were no tumor types that were mentioned and no description of the number of normal and tumor tissues involved, making it very difficult to evaluate their data.

We thank the reviewers for pointing this out. While the number of samples for each tumor type were included in Table 1 within the text of the methods sections, it had not been included in the results section. A brief description of the tumor types and non-tumor tissues used in our analyses as well as a reference to Table 1 has now been included in the results section.

2. A second major concern is the very limited tumor samples. Although it is well known that pediatric brain tumor tissues are very difficult to obtain, it is also very well established that these tumors are composed of many molecularly distinct subtypes. With a small collection of 32 tumor samples that included tumors of different locations, different pathological diagnosis, different grade (high and low), there is a lack of statistic power to draw any meaningful conclusions from this study.

We agree with the reviewer that the sample size is relatively small compared with analyses of other tumor types and is a limitation that we and others studying pediatric brain tumors contend with. However, for each tumor studied we include molecular information from hundreds to thousands of cells which addresses some potential concern with distinctions among subtypes. It is quite difficult to accrue a larger sample size for these tumor types. It is also very difficult to obtain non-tumor pediatric brain tissues to be used in research studies. Our sample size may be small, however, it is a starting point to begin to investigate the effects of heterogeneity so that larger studies in collaboration with other medical centers or consortiums can be conducted to increase statistical power. In comparison to the few published studies have investigated single cell states of pCNS tumors, here, we contribute the largest sample size (>40% increase in total pCNS samples available for analysis), and one of the largest number of nuclei analyzed (>74% increase in total pCNS nuclei available for analysis). For instance, Filbin et al analyzed ~2,500 cells in their study investigating H3K27M pediatric gliomas (Science, 2018), Hovestadt et al characterized ~9,900 cells from medulloblastomas (Nature, 2019), and Gillen et al assessed 18,500 cells from ependymomas (Cell Reports, 2020). Additionally, to conduct a study with larger sample sizes in which all samples would have single cell RNA-seq, bulk RNA-seq, epigenome-wide DNA methylation, and hydroxymethylation profiles, the cost would be extremely high and require fresh frozen specimens.

Minor concerns:

1. In Figure 2E, were the shared dhmcpgs (90 and 10/4%) also detected in normal brain tissues? What were differences among tumors of different locations?

In Figure 2E, the dhmcpgs indicates differentially hydroxymethylated CpGs in comparison to the non-tumor brain tissues which would mean the same patterns would not be detected in the non-tumor brain tissue. Due to the sample size, we were not able to stratify further among the different tumor types.

2. Were the normal tissues age matched and location matched?

We were only able to use two non-tumor brain tissue that had all single cell and bulk RNA-seq along with DNA methylation and hydroxymethylation profiles in our analyses. While the age range is with the pediatric CNS tumors, the non-tumor samples were both supratentorial.

3. On page 9, 1st paragraph, “Our results indicate that hydroxymethylation may be associated with changes in gene expression for certain genes in pediatric CNS tumors” does not mean much. It is hard to believe any conclusion can be drawn based on such loose correlations.

Thank you for pointing this out. While it is not a sweeping conclusion of most genes that feature this association, the objective of the statement was to highlight that dysregulation of transcription of certain genes that may be of importance in pediatric CNS tumors may be attributed to altered hydroxymethylation states. To convey in a clearer manner, we have revised the statement in the manuscript to the following:

“Our results suggest altered hydroxymethylation states in regulating gene expression of certain pediatric CNS tumor-associated genes, providing an avenue for further investigation”.

4. This is the first time to see someone using ACT as an abbreviation for astrocytoma.

For the purpose of brevity and clarity in our figures, we arbitrarily decided to use ATC for astrocytoma as it needed to cover broader types within astrocytomas.

5. This study emphasizes the “cell type-specific” cytosine modification, but the “progenitor-like cells were an aggregation of neural stem cells, radial glial cells, oligodendrocyte precursor cells and unipolar brush cells”, not strictly cell-type specific.

We thank the reviewer for pointing this out. While it would have been ideal to utilize more granular cell types, as CellIDMC method utilizes the cell types as variables in the interaction term of the model, there would have been too many variables for our sample size. The cell type-specificity level for our analyses in this manuscript is more general grouping (of neuronal-like cells and stemlike cells).

Reviewer #3 (Remarks to the Author): Expert in single-cell multi-omics and brain cancer genomics

In this manuscript, Kyung Lee et al. aimed to evaluate cell-type alterations in methylation (5mc and 5hmC) in a variety of pediatric tumour tissue relative to non-tumor tissue, as well as their association with gene expression level.

The findings related to 5-hmC are interesting. However, with bulk methylation data and limited CpG sites captured, the authors should be cautious in interpreting the results. Further, even though there are some novel findings from a biological standpoint, some of the statements and results are not adequately supported by quantitative/statistical measurements. Several of the findings must be interpreted with greater precision.

Overall, the paper would benefit greatly from a more detailed explanation of the rationale and scope of the analyses, a stronger and more detailed description of the methodology, and the addition of specifics to the legends and figure panels.

The following points should be clarified and edited in the study:

1. Have the authors considered comparing their results to other single-cell sequencing studies on pediatric CNS tumors? This is a small cohort, so it would be very important that this small cohort is representative of larger patient cohorts but not a collection of extreme cases.

We thank the reviewer for the suggestion. In comparison to the few published studies have investigated single cell states of pCNS tumors, here, we contribute the largest sample size (>40% increase in total pCNS samples available for analysis), and one of the largest number of nuclei analyzed (>74% increase in total pCNS nuclei available for analysis). For instance, Filbin et al analyzed ~2,500 cells in their study investigating H3K27M pediatric gliomas (Science, 2018), Hovestadt et al characterized ~9,900 cells from medulloblastomas (Nature, 2019), and Gillen et al assessed 18,500 cells from ependymomas (Cell Reports, 2020). We observed similar cell types compared to the other limited studies. For example, previous studies have shown major prevalence of radial glial cells in ependymomas as we have found in the ependymomas we assessed. In addition, other work in astrocytomas showed abundant oligodendrocyte precursor cells in astrocytoma that also were observed in the astrocytomas we assessed.

This work was submitted as a complementary manuscript to another manuscript that focused on the cell type and transcriptomic heterogeneity of these same tumors. The other manuscript also is in revision for further consideration at Nature Communications. We have included the above comparisons in that manuscript as it was more relevant. We agree with the reviewer that these comparisons are important references. While it was mentioned more in detail in the other manuscript, we have updated this manuscript to include a reference to these comparisons (page 6).

2. Could the authors provide more details on the implementation of CellIDMC?

CellIDMC (Zheng et al, 2018; PMID: 30504870) is a statistical method to determine both differentially methylated CpGs and the cell type driving the differential methylation. In our study, we utilized CellIDMC to identify cell type specific differential hydroxymethylation and methylation in the pediatric CNS tumors compared to non-tumor pediatric brain tissue. Separate models were run for each tumor type with the same cell types. We chose categorized the cell types present in our samples to two broader categories (neuronal-like cell types and progenitor-like cell types) as CellIDMC incorporates the cell type proportions as interaction terms in the regression model. Having more cell types would have been too many terms in our model for the sample size we had in our study. We now include a more detailed description of CellIDMC and its

implementation in the methods section (pg. 9) of the manuscript.

3. The authors should consider validating the results by sorting the cell type and measure the methylation, or using single cell methylation assays?

Thank you for the suggestion. While single cell methylation assays would certainly be of interest, protocols and approaches for single cell methylation assays are quite nascent, require substantial substrate, and are not yet standardized. Moreover, it would have been difficult to utilize flow activated cell sorting from surface markers as our samples were kept as fresh frozen samples for decades. Isolating nuclei for single/bulk RNA-seq and DNA for methylation assays would be very difficult. Specific nuclei-level markers for various cell types in the brain are limited as well.

4. Is this the first study used different methods to identify cell type-specific alterations to cytosine modifications in pediatric CNS tumors? If not, discussion should incorporate the result from some previous research. In Results, it was briefly mentioned that “it was quite unexpected that the observed differences were almost solely in hydroxymethylation and not in methylation”.

To our knowledge, our study is the only one to identify cell type-specific alterations to cytosine modifications in pediatric CNS tumors. We were surprised by the extent of increased tumor CpG hydroxymethylation when compared with non-tumor brain tissue. The observed increase in 5-hmC in the tumors compared to non-tumor tissue may be indicative of a more predominant role of progenitor cell types in the pediatric CNS tumors relative to adult CNS tumors.

There are widely accepted/clinically applied DNA methylation-based methods for diagnostic classification of CNS tumors that may be impacted by the incorporation of 5hmC measures. We had expected that higher alterations in methylation to exist in comparison to non-tumor tissues due to its ability to distinguish among tumor types. .

5. Only two non-tumor tissue were included in the study, which is rather few. Including at least another non-tumor tissue would strengthen the conclusions presented in the manuscript.

We agree that having only two non-tumor tissues is a limitation. However, we only wanted to use non-tumor samples in which we had matching single cell RNA-seq and methylation/hydroxymethylation data for which led us to subset to these two samples. It is difficult to obtain non-tumor pediatric brain tissue which limited our analyses. Although our non-tumor sample size is very limited, we thought it was critical to investigate direct comparisons between pediatric CNS tumors and non-tumor pediatric brain tissue as it has not yet been explored. For these tumor types, this is the first investigation to utilize an integrative single nuclei RNA-seq and hydroxymethylation profiles of non-tumor pediatric brain tissue to our knowledge.

6. The Methods would benefit from a more detailed description. The provided information is not enough to reproduce the results reported in this manuscript.

Thank you for pointing this out. We had supplemented more references to single cell RNA-seq and transcriptomic level specific manuscript and have added additional details into the methods section of this manuscript.

7. The analytical codes used to generate the main results of the paper should be provided.

Thank you for pointing this out. We have added a GitHub link for code used in this analysis.

8. It would be beneficial to include details on number and type of pediatric CNS tumor samples upfront, for instance on page 5, when first introducing the results. This would give the reader a better understanding on the cohort of sample types available for analyses upfront.

We thank the reviewer for this suggestion. We have included a brief description of number of samples used per tumor/non-tumor tissue at the beginning of the results section in page 5.

Along this line, reference to Supplementary Table 1 is missing from the main text. Further, number of samples reported in Table 1 differs compared to Supplementary Table 1 (see number of Astrocytoma samples). In the Methods section, it is mentioned that 32 samples were measured, but 33 samples were treated with DNA methylation and hydromethylation assays.

Thank you for pointing this out. We have revised the number of astrocytomas from 7 to 8 for a total of 34 samples in our analyses.

9. Figure 1A is not very intuitive (median beta values of 5-mC have higher proportions around 0 - 0.1 and 0.8 - 1, while median beta values of 5-hmC fall into 0 - 0.2?).

What is the statistical significance of tumor vs. non-tumor tissue?

May also consider switch the order of the legend? (i.e., 5-mC tumor & non-tumor, 5hmC tumor & non-tumor).

It has been previously established that 5-hmC is present at much lower levels than 5-mC. Our results follow previously published findings, in which the levels for 5-hmC ranges from 0 – 0.2 and levels for 5-mC ranges from 0.5 – 1. While we did not initially report results from a statistical test comparing tumor to non-tumor tissue, we have added results from the Kolmogorov-Smirnov tests for each 5-mC and 5-hmC. The results indicate significantly different distribution of 5-mC and 5-hmC in the tumor tissues and non-tumor tissue under $p < 0.05$ threshold. We thank the reviewer for the suggestion in the change of the legend and have revised the figure to reflect that.

10. In figure 1B and D, a visualization that includes inter-quartile ranges (e.g., boxplots) would make the assessment of outlier samples easier to detect and identify for the reader. How did the authors identify outliers to be removed in the analysis of Supplemental Figure 2?

We thank the reviewer for the suggestion. We have added boxplots to Figures 1B and 1D. The outliers were identified by Grubb's test for outliers under the significance threshold of $p < 0.05$.

11. In figure 1C and Supplementary Figure 2C, it would be beneficial to color code the dots by tumor type for easier assessment of the results. Also, $R = 0.44$ is a little bit concerning. Most of the data points for 5-hmC MDI reside around 0.04 while 5mC MDI from 0.04 to 0.12.

We have revised Figure 1C and Supplementary Figure 2C to color each point by tumor type. We reported the Spearman correlation coefficient which resulted in a slightly weaker correlation coefficient than Pearson correlation. While it is not exceptionally

strong correlation coefficient, there appears to be an incremental relationship between 5-hmC MDI and 5-mC MDI.

12. The authors should better contextualize the results in Figure 1A. In cancer, generally, it is common to observe globally decreased methylation in tumor tissue compared with normal, with a focally increased methylation of promoters or CpG islands. It would be interesting to repeat analyses of Figure 1 to also report 5-mC, 5-hmC and MDI scores at specific genomic regions to compare to epigenome-wide scores.

We thank the reviewer for the suggestion. We have determined MDI for specific genomic contexts (promoter, enhancer, gene body, and exon) and have found similar patterns as for overall MDI for which we have added the plots as Supplementary Figure 3 and have added the following text:

In addition, we determined MDI for distinct genomic contexts and again found consistent results in which 5-mC MDI, but not 5-hmC MDI values significantly varied among tumor types (Supplementary Figure 3). Interestingly, both 5-hmC MDI and 5-mC MDI in gene body, enhancer and exon regions were slightly, but statistically significantly higher than 5-hmC MDI and 5-MDI when adjusted for tumor types and grade (Supplementary Table 3. For both 5-hmC and 5-mC, MDI were highest in enhancers, then gene body/exon regions and were lowest in promoter CpGs.

13. The identification of the cell type composition of pediatric CNS tumor tissue and non-tumor pediatric brain tissue is an important component of the analysis; however, methods on how the cell types are identified using single-cell nuclei RNA sequencing is lacking. In general, a more in-depth description of the Methods would help the work to be reproduced. Are the cell types common across all these pediatric CNS tumor tissues? Or each pediatric tumor type is expected to have its own cell type labels?

Thank you for pointing this out. Cell types from single nuclei-RNA seq were identified in a manuscript submitted concurrently to this journal with this current manuscript which is in revision. Briefly, cell types were identified by, first, classical markers for different cell types in the brain. For example, markers like *GFAP* and *AQP4* for astrocytes and *MOG* and *PLP1* for oligodendrocytes were used. Cell types were then validated by gene set enrichment testing for genes known to be enriched in specific cell types using Variance-adjusted Mahalanobis (VAM) method (Frost, 2020; PMID: 32633778). We have revised to indicate the methods in this current manuscript in pg. 17. Full description of cell type identification can be found in Lee et al.

14. “Based on the cell type proportion distributions for all of our samples, we identified neuronal-like cells (NEU), neural stem cells (NSC), oligodendrocyte precursor cells (OPC), radial glial cells (RGC), and unipolar brush cells (UBC) as having the most variance”.

It is unclear what is the scope and rationale behind the analysis of Supplementary Figure 4. From Supplementary Figure 4A, it is also unclear how the authors can state that UBC and NSC have the most variance when they are centered around 0 on PC1 and PC2 axes. Is this pseudo-bulk with one data point representing one cell type from snRNA-seq?

From Supplementary 4A, it shows only OPC, NEU, and RGC as having the most variance; UBC

and NSC are clustering with the rest of cell types. Again, in general, the method and approach implemented to obtain this specific result are unclear.

It would be interesting to compare the "degree of heterogeneity" between the single-cell sequencing datasets as a more appropriate measure of "variance". Ecological measures like Shannon diversity would work great in assessing whether a given tumor type is more heterogeneous vs. others.

We thank the reviewer for the suggestion. These cell types were chosen from the single nuclei RNA-seq data. We took into account the inherent nature of the cell type (i.e. stemness level of the cell type – identified in paper #1), PCA analyses and distribution of proportion for each tumor types to select cell types in the CellDMC analyses. The variances were assessed to select cell types that contributed major variability among the tumor types and with the non-tumor tissue. Therefore, analyses for degree of heterogeneity were not conducted for the analyses. While we agree that utilizing measures like Shannon diversity would be important to assess overall heterogeneity of the tumors, to reduce the cell type effects in our analysis we selected the most important cell types that could confound identification of molecular differences. We have added this objective of this analysis into pg. 11 of the manuscript.

15. Supplementary Figure 4 legends is not reflecting the panels. Missing panel B or C description in the legends and Supplementary Figure 4C is not referenced in the main text. Supplementary 4B looks like clustering of cell types (with two PCs in the axis). The legends sound like proportions or differences? Not very clear what is done from S4A to S4B. (It is likely that they are referring to S4C as S4B currently.)

We thank the reviewer for pointing this out. Text referencing these figures and Supplementary Figure 4 figure legends have been revised to describe the figures.

16. "For each tumor type we compared proportions of cell types with non-tumor pediatric brain tissue. Supporting our principal component analysis, the cell types with the greatest differences were NEU, NSC, OPC, RGC, and UBC".

From supplementary Figure 4C, most of the p-values are not significant (likely due to lack of power due to limited sample size). The relevance of this result is unclear. The cell types highlighted by the authors seem to have a strong tumor-type-specific enrichment. NEU is mainly present only in NT. NSC is mainly present (at low levels) in EMB. OPC is mainly present in ATC. RGC is mainly present in EPN. UBC is only present in EMB. The authors should better clarify the methods, approaches, rationale, scope and better describe the relevance and meaning of the results obtained.

We agree with the reviewer that the p-values are not statistically significant and that cell types were tumor type specific. However, the purpose of the figure was not to claim significance of the differences of cell type proportions among the tumor types but to display which cell types were most variable to select the cell types to adjust for in the CellDMC model for our epigenome wide association study. While the cell types in the progenitor-like cell types are tumor type-specific, the models were run separately by tumor type. By aggregating the progenitor-like cell types together, we were able to adjust for the major cell types present for each tumor type, but also account for the cell types

present at smaller proportions. We have added more explanation of the rationale behind the PCA and comparisons of the cell type proportions in pg. 6 – 7 of the manuscript text.

17. “When all five cell types (NEU, NSC, OPC, RGC, and UBC) were incorporated into the model, we observed low number of dmCpGs associated with each tumor type.”

Even when the model is not adjusted, lower number of dmCpGs is observed?

If such minor difference is observed between the 5mc status, does that suggest the previously reported dmCpGs is likely only associated with 5hmc?

Then, this is contradicting with the claim in the introduction “it has been indicated only 5-methylcytosine signal from oxidative bisulfite-treated DNA alters the classification”. Please clarify.

We observed nearly universal decreases in the number of significantly differentially modified CpG sites in tumor compared with nontumor when adjusting for cell type variation as compared with unadjusted models. There were slightly higher numbers of differentially 5-mC for EMB tumors in the fully adjusted model (202 CpGs, 0.04%) compared with unadjusted (183 CpGs, 0.04%) (**Table 3**). Identifying fewer significantly differentially modified CpG sites when adjusting for cell type was not unexpected.

However, we were surprised to observe, for all tumor types, the scope of significantly differentially hydroxymethylated CpGs was much higher than differential 5mC. **Measures of 5-mC with bisulfite conversion cannot distinguish between 5-mC and 5-hmC and such measures have been used to build CNS tumor subtype classification. Tandem treatment and measures with bisulfite conversion and oxidative bisulfite conversion are needed to determine unmodified cytosine, methylated cytosine and hydroxymethylcytosine. In addition, tandem treatment approaches require fresh frozen tissue.** Relative to studies that only measure 5-mC, there has been very limited investigation of 5hm-C modifications. Our findings underscore the importance of 5hmC in CNS tumorigenesis, and suggest that prior work identifying sites with differential 5mC would have observed

1. different sets of CpGs if adjusting for cell type variation, and
2. a substantial fraction of CpGs with differential hydroxymethylation (not 5mC) if they had used tandem conversion and measures
3. different tumor classifications if measures of 5mC and 5hmC had been conducted

Table 3.

	Unadjusted model dmCpGs N (%)	dmCpGs that are also dhmCpGs N (%)	Adjusted model dmCpGs N (%)	dmCpGs that are also dhmCpGs N (%)
Astrocytoma (ATC)	7 (0.001%)	3 (42.9%)	0 (0%)	0 (0%)
Embryonal (EMB)	183 (0.04%)	90 (49.1%)	202 (0.04%)	15 (7.4%)
Ependymoma (EPN)	39 (0.008%)	25 (64.1%)	0 (0%)	0 (0%)
Glioneuronal/neuronal (GNN)	3 (0.0006%)	1 (33.3%)	0 (0%)	0 (0%)

18. In the analysis of Figure 2, how is the cell-type specificity (see Supplementary Figure 4C) considered? The number of differentially hydroxy(methylated) CpGs would likely differ as a function of a given cell type? How is the different number of samples considered in the analysis (e.g., only two non-tumor brain tissue)?

As specific cell types have been shown to have distinct 5-mC and 5-hmC signatures, the decrease in number of dhmCpGs and dmCpGs associated with tumors was expected because of the cell type composition differences between tumors and non-tumor tissues. While small number of non-tumor tissue used in the comparison is a limitation, we expect a similar decrease in dhmCpGs and dmCpGs associated with tumors due to the cell type specific nature of 5-hmC and 5-mC.

Could the authors compute a differential analysis of a given cell type between tumor and non-tumor tissue to overcome the cell type composition confounding factor? It would be interesting to repeat this analysis by focusing on a specific genomic region, e.g., promoters or CpG islands to assess whether these results are different compared to adopting a genome-wide approach.

The Illumina EPIC methylation array that we used in this manuscript covers ~800,000 CpGs across the genome which includes different regions of the genome like promoters or CpG islands. If we were to do a focused analysis selecting CpGs in specific regions that had already been included in our analysis, the model coefficient estimates and P-values would be the same, but the corrected P-values would be lower. From the reviewer's suggestions, we assessed the proportion of CpGs at each genomic context within each model. We tested the relationship between proportion of dhmCpGs per genomic context and number of cell types included in the model with a Spearman rank test. We found that the proportion of dhmCpGs in promoter regions within 200bps of the transcription start site (TSS200) and 1st exons were positively associated with increased number of cell types included in the model (Supplementary Figure 10A, Figure 2F). Moreover, we found negative relationship between # of cell types included in the model and proportion of dhmCpGs in the gene body, open chromatin and transcription factor binding sites. We now include text describing these results on page 6.

Can the authors provide more details on the 28 dhmCpGs that were found in common across all samples?

We have included Supplementary Table 3A – C and additional text to include additional details on the 28 shared dhmCpGs as follows:

The 28 shared CpGs were located predominantly in island (42.9%) and open sea (42.9%) regions in relation to CpG islands (Supplementary Table 3B). In addition, 64.3% of the shared dhmCpGs were located in DNase hypersensitive sites (Supplementary Table 3C). The shared CpGs were located in genes like ESRRG, HECA, THBD, and TJP1 (Supplementary Table 3A).

Overall, as for Figure 1, the rationale of the analyses reported in Figure 2 are not well laid out. What is the mere number of differentially CpGs tell us in terms of advancing biological knowledge of pediatric tumor tissue vs. non-tumor tissue?

While many studies have described the genomic burden, or a lack thereof, in pediatric CNS tumors, limited studies have outlined epigenomic burden in these tumors. Even when the epigenome has been investigated, cell type compositions were not considered. With this analysis, we were able to not only identify epigenomic alterations of *both* 5-hmC and 5-mC in pediatric CNS tumors, but we also were able to describe the effects of cell type composition on altered cytosine modifications. Figure 2 identifies CpGs associated specifically with tumors, and not cell types. Our results provide a clearer picture of the molecular alterations in these tumors for potential therapies or markers for tumors.

19. For the analysis of bulk RNA-seq differential gene expression, the authors report that the majority of differentially expressed genes were increased in expression compared to the non-tumor pediatric brain tissue. better specify the cutoff: $-\log_{10}(p\text{-value}) = 2$? Did not find any information in the figure legend.

Also, could this reflect the imbalanced dataset, given that only 2 non-tumor tissue are available? Have the authors tried to repeat the analysis by equalizing sample size to see whether this finding holds true?

What do the findings of “keratinization” vs. “insulin receptor signaling” mean biologically and in the context of pediatric tumor tissue?

Thank you for pointing this out. We have added the adjusted p-value < 0.05 threshold into the supplementary figure legend.

While the number of our sample size is a limitation, acquisition of pediatric non-tumor brain tissue is difficult. Moreover, as we only included samples collected within our cohort with matching single nuclei RNA-seq, bulk RNA-seq and both DNA methylation and hydroxymethylation profiles, publicly available datasets with only single -omic measures could not be included in our analysis.

We sought to include the pathways associated with the shared differentially expressed genes to begin to understand shared pathways that may be dysregulated during tumorigenesis and to highlight pathways of interest for developing therapy in pediatric CNS tumors. As we reduced potential confounding effects from cell composition differences, we believe the identified pathways are more closely associated with carcinogenesis rather than specific cell types. As such, genes along the insulin receptor signaling pathway may be of appropriate targets for therapy. We now have included more rationale in the manuscript text on page....

20. It is unclear how the authors selected the genes to be reported in Figure 3. As already mentioned for Figure 1 and Figure 2, the lack of detail in describing the analysis and methods make it difficult to assess and interpret the robustness of the results.

We regret the lack of clarity for gene selection, we now include a reference to the differential expression analysis from which we selected the genes for what is reported in Figure 3. Specifically, the genes that were labeled in the Figure 3 plots were genes that were identified to be differentially expressed in the tumors based on the differential gene expression analysis reported in the first paragraph on page 9 (differential expression analysis results shown in Supplementary Figure 9).

21. A more quantitative approach to support the statements related to Figure 3 is needed. The authors should quantify whether there is indeed a preferential enrichment of genes that are both highly expressed and show negative 5hmC beta coefficient. Many of the statements related to Figure 3 are vague and lack direct quantification using the methylation and expression data available.

Unfortunately, the enrichment tests for three tumor types (ATC, EPN, and GNN) would not be appropriate due to the small number of CpGs that would be included in the analysis. Sufficient differentially hydroxymethylated CpGs were identified for embryonal tumors to conduct an enrichment testing, and we report those results.

22. It is unclear how the cell-type specific differential CpGs are found in the analysis of Figure 4. What are the group comparisons to determine what is decreased vs increased? Are the authors comparing the same cell type (e.g., neuronal-like) between tumor and non-tumor tissue? Once again, more detail in the Methods, main text and Legends would greatly help the reader in understanding the results and findings.

Each tumor type was compared to non-tumor tissues in separate CellDMC models that included the same cell types in the models. CellDMC incorporate cell type proportions as interaction terms to identify cell type specific differential CpGs. The model identifies which cell type is driving the change in 5-hmC or 5-mC in tumors compared to non-tumor. We have revised the text and have included more specific details of what is being compared in the main text. We also have revised the methods section including more detail on the CellDMC approach (originally reported in Nature Methods in 2018, PMID 30504870), as:

Cell type-specific differential hydroxymethylation and methylation for each tumor type were identified using CellDMC¹⁰³. CellDMC is a statistical model that identifies both differentially methylated CpGs and which cell type drives the differential methylation by incorporating cell type proportions as interaction terms in the linear regression model in the epigenome wide association study¹⁰³. Proportions of cell types of interest (neurons and progenitor-like cell types) were pulled from the single nuclei RNA-seq dataset. To limit overfitting the model in our relatively smaller sample size, we aggregated the progenitor-like cell types into a single cell type category. The progenitor-like cell types included neural stem cells (NSC), radial glial cells (RGC), oligodendrocyte precursor cells (OPC), and unipolar brush cells (UBC). UBCs were included due to the high levels of stemness score in the cell types identified previously. Separate models to compare each tumor type to the non-tumor tissue were run with the same cell types (progenitor-like and neuronal-like cell types) included in each model.

23. The analysis of Figure 4 seems to be confounded or mainly driven by the number of cells/nuclei observed in each tumor type. For instance, it is not surprising to see that the >50% of cell-type differential CpGs for the progenitor-like cell types are from EMB and EPN, being the two tumor with the highest number of progenitor-like cells as reported in Supplementary Table 4.

This same concern applies to Supplementary Figure 20 given the big imbalance between number of neuronal-like vs. progenitor-like cell types. What is the p-value of the OR reported in Figure 5?

Overall, the data presented should be revised to support the claim that “most of the hydroxymethylation alterations occur in the progenitor-like cell types and are tumor-type-specific”.

While the results shown may be intuitive, there have not been studies that have specifically shown both distribution of cell types and specific cell types driving differential hydroxymethylation and methylation. Had we only made comparisons of tumors to non-tumor samples without considering cell type proportions, results would have been confounded by cell type since the epigenome profiles of the tumors in our cohort are from bulk tissue (and methods for single cell cytosine modification measures are not generally available). The integrated analyses presented here enable a more granular investigation identifying cell type-specific epigenomic alterations. The results from our analyses reduce confounding from cell-type-specific epigenomic profiles.

We did not show the p-value in Figure 5 as we reported the 95% confidence intervals in the plot. We have added Supplementary Table 7 with the odds ratio, 95% confidence interval and p-value.

To address that there were more dhmcpgs driven by neuronal-like cell types in astrocytoma we have revised the conclusion as follows:

Our findings indicate that hydroxymethylation alterations are driven by different cell types in different tumor types.

24. It would be good to use a score or statistics in Figure 6B to show the enrichment.

Because of the dhmcpgs for *HDAC4* and *IGF1R* were only focused in on certain genomic contexts, we have revised the figure to show proportion of CpGs that were differentially methylated out the CpGs in specific contexts of the *HDAC4* and *IGF1R*.

25. Lack of statistical testing in Figure 6D makes it difficult to support the claim that the progenitor-like cell types had higher levels of HDAC4 expression and that IGF1R gene expression levels were higher in the progenitor-like cell types.

All in all, the results reported as is in Figure 6 do not strongly support the claim that “Our results suggest potentially critical roles of hydroxymethylation of CpGs located within the gene body regions in regulating the gene expression of critical cancer genes, like HDAC4 and IGF1R”.

We agree with the reviewer that it is not a strong correlation. However, it is not often the case that strong correlations between changes in DNA methylation and hydroxymethylation and changes in gene expression are observed. It is more likely that there is a nuanced association between the two which our results suggest. We have revised to indicate there seems to be an observable, yet not strong, relationship between the two in the manuscript text.

In Figure 6D, the authors reported expression for “other glioma”. The samples in this category need to be specified.

We want to thank the reviewer for pointing this out. We have removed the other glioma in Figure 6D as those samples were not part of the analysis for this manuscript.

26. “Our results suggest that pediatric CNS tumors may be characterized by non-mutational epigenomic reprogramming more so than genomic aberrations”. None of the analyses reported in this manuscript included genomic aberrations so this statement should be toned down or omitted.

Thank you for pointing this out. It was included as we have another manuscript associated with this manuscript suggesting this. However, to keep in line with what is presented in the current manuscript, we have removed ‘more so than genomic aberrations’ in the manuscript text.

27. “Pediatric brain cancers have been shown to contain somatic mutations in epigenetic regulator genes such as H3F3A, KDM6A, and MLL3”. Are any other data type available from these patient samples? If genomic information is available, the authors could/should investigate it to correlate genomic aberrations with the observed epigenetic aberrations.

We thank the reviewer for this suggestion. We chose not to include the genomic information in this current manuscript as we have included genomic information of these samples in another manuscript that has been reviewed at Nature Communications and is in revision. We have included a reference to the other manuscript for genomic characterization we identified. In our samples, we did not observe enough genomic alterations in epigenome-related genes to determine relationships to the epigenomic alterations.

28. The authors should provide additional supplementary tables related to the differential CpGs analysis and differential gene expression (e.g., the mentioned 702 genes, etc...).

Thank you for pointing this out. We have included additional supplementary tables of the results. Due to the large file sizes, we have only included the significant CpGs for each of the models and the tumor types.

29. Overall, figures and figure legends lack detail and specificity (e.g., how many samples? How many cells? how many genes?).

Thank you for pointing this out. Similar to the genetic information, the specific cell numbers, etc. were not included in this manuscript as they are being reviewed in another manuscript at the same time. As the other manuscript has also been reviewed, we have included more references for additional detail and have also revised this current manuscript to include more information.

REVIEWER COMMENTS

Reviewer #1 (Remarks to the Author):

The authors made some effort in their responses to my comment. There are, however, rooms for improvement:

1. The major concern #1 was referring to the whole section of RESULTS, not just one simple “brief description”. Tumor sample information and the actual numbers of each tumor type should be described in every significant findings. This is very important.
2. For major concern #2, there was not discussed in the main manuscript text. The response (justification) was not strong. The limitations of their small sample size have to be discussed, and critically analyzed.
3. For minor concern #1 (Figure 2E) and #5 “cell type-specific” cytosine modification, the authors need to present the “we were not able to stratify....” and “...too many variables” in the text and fully acknowledge the limitation of their study.

Reviewer #3 (Remarks to the Author):

In general, the authors have effectively tackled all the reservations I previously had. Nevertheless, a few adjustments are still needed before I can endorse the manuscript for acceptance.

Regarding my observations about statistics, some statements and findings still lack adequate backing through quantitative or statistical measures. Certain discoveries require a more precise interpretation, and caution should be exercised in drawing overall conclusions due to either the absence of direct statistical testing (e.g., Figure 6D) or the limited statistical power resulting from a small sample size.

Another area of concern pertains to the Methods section. Adding a few more details would be beneficial to ensure the complete clarity and reproducibility of the results. For instance, the authors utilized MULTI-Seq to combine snRNA-seq samples. The authors should provide specifics (including code) on how to separate the pooled samples back into individual samples. This is just one example. Despite the

reference to a concurrently revised manuscript in the same journal (Nature Communications), this should not discourage the provision of necessary information for the scientific community to replicate the outcomes achieved in this paper.

REVIEWER COMMENTS

Reviewer #1 (Remarks to the Author):

The authors made some effort in their responses to my comment. There are, however, rooms for improvement:

1. The major concern #1 was referring to the whole section of RESULTS, not just one simple “brief description”. Tumor sample information and the actual numbers of each tumor type should be described in every significant findings. This is very important.

Following the reviewer’s feedback, we had included the number of samples for each tumor type (8 astrocytoma, 6 embryonal tumors, 10 ependymoma, 8 glioneuronal/neuronal tumors) and the 2 non-tumor tissue in the prior revision of the results section of the text (pg. 5). These set of samples for each tumor type and non-tumor tissue were used for all following analyses in the manuscript. Regrettably, some of the sample specific points were hidden behind the boxplot in Figure panels 1B, 1D, Supplementary Figure 2A, 2B, as well as Supplementary Figure 3. We have revised the figures to clearly indicate the sample specific points in these figures.

Additionally, we now include sample size per tumor type in several places throughout the results section of manuscript including the figure legends:

(pg 5): Tumor tissues (N = 32) displayed a decrease in median 5-hmC beta values and a slight increase in median 5-mC beta values compared to non-tumor tissue (Non-tumor tissue N = 2; KS test: 5-mC: $D = 0.019$, $p < 2.2e-16$; 5-hmC: $D = 0.19$, $p < 2.2e-16$; Figure 1A).

Figure 1. Global methylation dysregulation, but not global hydroxymethylation dysregulation, is associated with tumor type and grade. A) Cumulative proportion of 5-hmC and 5-mC in tumors (N = 32) and non-tumor tissue (N = 2). B) Methylation dysregulation index of 5-hmC and 5-mC by tumor type (N = 8 (ATC), 6 (EMB), 10 (EPN), 8 (GNN)) and D) grade (N = 14 (G1), 5 (G2), 6 (G3), 6 (G4)). Differences in MDI were calculated using the Kruskal-Wallis test. C) Correlation between 5-hmC MDI and 5-mC MDI calculated using Spearman rank correlation. Linear regression line is indicated by the blue line. 95% confidence interval indicated by gray bands. Color of each point indicates tumor type.

(pg. 7): We conducted an epigenome-wide association study to determine the differential hydroxymethylated and methylated CpGs associated with each tumor type (N = 8 (ATC), 6 (EMB), 10 (EPN), 8 (GNN)) compared to non-tumor pediatric brain tissue (N = 2).

(pg. 8): We then compared transcriptome data from bulk RNA-seq in each of the tumor types (N = 8 (ATC), 6 (EMB), 10 (EPN), 8 (GNN)) with non-tumor pediatric brain tissue (N = 2).

(pg. 9): To identify potentially important gene regulation by differential hydroxymethylation we compared changes in hydroxymethylation in dhmCpGs from the five-cell type-adjusted model with gene expression in each tumor type (N = 8 (ATC), 6 (EMB), 10 (EPN), 8 (GNN)).

2. For major concern #2, there was not discussed in the main manuscript text. The response (justification) was not strong. The limitations of their small sample size have to be discussed, and critically analyzed.

We thank you for the feedback. We have added the following to the manuscript to incorporate and discuss the limitations from sample size:

(pg. 13): However, these differences in gene expression in each cell type of each tumor type compared to the same cell types in non-tumor tissues were not statistically significant which was likely due limitations from sample size (Figure 6D).

(pg. 14): However as statistical significance levels were not reached in cell type specific differences in gene expression levels likely due to limited sample size, further experimentation is needed to validate these results.

(pg. 16): Accruing a large sample size for pediatric CNS tumors is extremely difficult as they are very rare in the general population. The limited sample size prevented us from including other potential variables like tumor location. As different parts of the brain may be composed of differing cell types, not adjusting for tumor location introduces limitations in our conclusions. However, as we compare the epigenome within major cell types, we believe that some limitations of not including tumor location were addressed. Furthermore, the limited sample size reduced our statistical power in our analyses. While our study does incorporate a reasonable sample size for these rare tumors, the smaller sample size limited the inclusion of other variables and cell types that may affect methylation and transcription into our models. Moreover, our study incorporates multiple genome-wide and epigenome-wide molecular features of the matched tumor sample to give a more comprehensive landscape of each tumor type. Multi-omic approaches involving single nuclei RNA-seq, bulk RNA-seq, 5-mC, 5-hmC epigenome profiles of different pediatric CNS tumors have not yet been investigated to our knowledge.

3. For minor concern #1 (Figure 2E) and #5 “cell type-specific” cytosine modification, the authors need to present the “we were not able to stratify...” and “...too many variables” in the text and fully acknowledge the limitation of their study.

Thank you for the suggestion. We have revised the following in the text:

(pg. 7): Age at diagnosis, sex and tumor purity were adjusted to reduce potential confounding from these variables in these linear models. Due to sample size, tumor location was not included in the model.

(pg. 16): The limited sample size prevented us from including other potential variables like tumor location. As different parts of the brain may be composed of differing cell types, not adjusting for tumor location introduces limitations in our conclusions. However, as we compare the epigenome within major cell types, we believe that some limitations of not including tumor location were addressed.

(pg. 20): Due to sample size, tumor location was not adjusted for in the linear regression models.

Reviewer #3 (Remarks to the Author):

In general, the authors have effectively tackled all the reservations I previously had. Nevertheless, a few adjustments are still needed before I can endorse the manuscript for acceptance.

Regarding my observations about statistics, some statements and findings still lack adequate backing through quantitative or statistical measures. Certain discoveries require a more precise interpretation, and caution should be exercised in drawing overall conclusions due to either the absence of direct statistical testing (e.g., Figure 6D) or the limited statistical power resulting from a small sample size.

Thank you for the suggestions. We have incorporated additional statistical analysis in Figure 3 and Figure 6D. Following the results, we have added the following text to quantify our observations and state the limitations of our results.

(pg. 9): When correlations between changes in 5hmC and changes in gene expression were performed to assess any directional relationship, the correlation coefficients across all tumor types were non-existent and not statistically significant even for genes that had statistically significant changes in gene expression (R, p-value = -0.03, 0.93 (ATC); -0.02, 0.85 (EMB); 0.096, 0.86 (EPN); 0.39, 0.19 (GNN), Figure 3).

(pg. 10): Unlike dhmcpgs, magnitude of changes in 5mC levels were negatively associated with magnitude of changes in gene expression for genes that did not have statistically significant gene expression changes (R = -0.45, p-value = 0.029) and genes with statistically significant gene expression changes (R = -0.41, p = 0.0002, Supplementary Figure 16).

(pg. 13): However, these differences in gene expression in each cell type of each tumor type compared to the same cell types in non-tumor tissues were not statistically significant which was likely due limitations from sample size (Figure 6D).

(pg. 14): As with HDAC4, the differences between each cell type of each tumor type and same cell type of non-tumor tissues were also not statistically significant (Figure 6D).

(pg. 14): However as statistical significance levels were not reached in cell type specific differences in gene expression levels likely due to limited sample size, further experimentation is needed to validate these results.

(pg. 16): Accruing a large sample size for pediatric CNS tumors is extremely difficult as they are very rare in the general population. The limited sample size prevented us from including other potential variables like tumor location. As different parts of the brain may be composed of differing cell types, not adjusting for tumor location introduces limitations in our conclusions. However, as we compare the epigenome within major cell types, we believe that some limitations of not including tumor location were addressed. Furthermore, the limited sample size reduced our statistical power in our analyses. While our study does incorporate a reasonable sample size for these rare tumors, the smaller sample size limited the inclusion of other variables and cell types that may affect methylation and transcription into our models. Moreover, our study incorporates multiple genome-wide and epigenome-wide molecular features of the matched tumor sample to give a more comprehensive landscape of each tumor type. Multi-omic approaches involving single nuclei RNA-seq, bulk RNA-seq, 5-mC, 5-hmC epigenome profiles of different pediatric CNS tumors have not yet been investigated to our knowledge.

Another area of concern pertains to the Methods section. Adding a few more details would be beneficial to ensure the complete clarity and reproducibility of the results. For instance, the authors utilized MULTI-Seq to combine snRNA-seq samples. The authors should provide specifics (including code) on how to separate the pooled samples back into individual samples. This is just one example. Despite the reference to a concurrently revised manuscript in the same journal (Nature Communications), this should not discourage the provision of necessary information for the scientific community to replicate the outcomes achieved in this paper.

We thank the reviewer for the suggestions. We agree that it is important for this manuscript to be clear and complete with all necessary information to replicate the approaches. We have added details regarding the MULTI-seq protocol and the demultiplexing in the manuscript.

(pg. 17): Nuclei were isolated from fresh frozen tissue samples following the Nuclei Pure Prep nuclei isolation kit (Sigma-Aldrich, St. Louis, MO) with some modifications. The samples were first washed with PBS to remove extraneous OCT the samples were frozen in. The tissue was homogenized with both wide and narrow pestles submerged in 2.5mL of the lysis buffer in a Dounce homogenizer. The lysate mixed with 4.5mL 1.8M sucrose cushion were gently layered on top of the 2.5mL of 1.8M sucrose cushion in Beckman ultracentrifuge tubes. Samples were centrifuged for 45 min at 13,000 RPM at 4 °C in an ultracentrifuge.

(pg. 18): Pooled nuclei were demultiplexed by hashtag oligonucleotides using HTODemux function in Seurat v4. Pooled samples were also demultiplexed using Vireo, a genotype based demultiplexing method. We performed genetic demultiplexing analysis using genotype data following the methods described in Weber et al implemented in a Nextflow workflow. Briefly, bulk RNA-seq reads from each sample were mapped to the reference genome (GRCh38.p13) using STAR Pooled single-nuclei RNA-seq reads were mapped to the reference genome using STARsolo. Variants among the samples within each pool were identified and genotyped with bcftools mpileup using the mapped bulk reads. Individual cells were then genotyped only at the sites identified using the bulk RNA using cellsnp-lite (mode 1a). Cell genotypes were used to identify the sample of origin for each cell using Vireo. Code for the genetic demultiplexing workflow can be found at <https://github.com/AlexsLemonade/alsf-scPCA/tree/main/workflows/genetic-demux>.

To integrate the methods, we first used sample identity assigned from the hashtag oligonucleotides. If the nuclei were confidently assigned a sample, it was compared to the genotype-based sample assignment. Those that did not match the same sample were filtered out. If the nuclei were assigned as a doublet or to none of the samples, the nuclei were assigned to a sample based on the genotype-based approach. 84,700 nuclei with confident sample assignment were used in analysis.

[References for the above text are included in the manuscript text.]

REVIEWERS' COMMENTS

Reviewer #1 (Remarks to the Author):

The authors have addressed my concerns and provided important revisions in the manuscript.

Reviewer #3 (Remarks to the Author):

In general, the authors have effectively tackled all the reservations I previously had in the second revision. I do not have additional comments.